# Characterizing four decades of accelerated glacial mass loss in the West Nyainqentanglha Range of the Tibetan Plateau

Shuhong Wang[1,2,3], Jintao Liu[1,2]*, Hamish D. Pritchard [3], Linghong Ke[2], Xiao Qiao[1,2], Jie Zhang[1,2], Weihua Xiao[4], Yuyan Zhou[4]

[1]State Key Laboratory of Hydrology-Water Resources and Hydraulic Engineering, Hohai University, Nanjing 210098, People's Republic of China.

[2]College of Hydrology and Water Resources, Hohai University, Nanjing 210098, People's Republic of China.

[3]British Antarctic Survey, Cambridge, CB3 0ET, UK.

[4]State Key Laboratory of Simulation and Regulation of Water Cycle in River Basin, China Institute of Water Resources and Hydropower Research, Beijing 100038, China.

*Correspondence: jtliu@hhu.edu.cn; Tel.: +86-025-83787803

**Abstract:**

Glacier retreat is altering the water regime of the Tibetan Plateau (TP) as the region's climate changes, but there remain substantial gaps in our knowledge of recent glacier loss in this region due to the difficulty of making direct high-mountain observations and this limits our ability to predict the future of this important water resource. Here, we assessed 44 years of glacier area and volume changes in the major West Nyainqentanglha Range (WNR) that supplies meltwater to the densely populated Lhasa River basin and Nam Co, the second largest endorheic lake on the TP. Between the two periods 1976-2000 and 2000-2020, we found that the glacier areal retreat rate was more than doubled (from -0.54 ± 0.21 % a⁻¹ to -1.17 ± 0.30 % a⁻¹) and surface lowering also accelerated ( from -0.26 ± 0.09 m w.e.a⁻¹ to -0.37 ± 0.15 m w.e.a⁻¹) with particularly intense melting after 2014. This acceleration is similar in both timing and magnitude to that observed for Himalayas glaciers farther south. Besides, the areal retreat rate and mass loss rate of most glaciers in WNR were not synchronized. To understand the sensitivity of WNR glaciers to climate forcing, we examined the effects of topography, debris-cover, and the presence of proglacial lakes on our observed changes. We found consistently faster areal retreat but slower thinning rates on steeper slopes and an inconsistent relationship with aspect. We concluded that our observed spatial and temporal patterns of glacier change were dominated by observed local variations temperature and precipitation, the melt-reducing role of supraglacial debris, and the increasing influence of ice-marginal lakes on glacier retreat.

## 1. Introduction

The Tibetan Plateau (TP) known as the "Water Tower of Asia", is the source of several of Asia's major rivers (Bolch et al., 2010) and glacial melt on the TP plays an important role in water supply for downstream populations, agriculture and industries along these rivers (Pritchard, 2019; Viviroli et al., 2007). Climate change over recent decades has boosted river discharge by increasing runoff from shrinking glaciers (Lin et al., 2020; Yao et al., 2007; Zhang et al., 2011a), but this boost

will eventually decrease as glacier area declines (Zhao et al., 2019). The sensitivity of ice loss to climate change is variable, however, and often poorly known, being a function of glacier size, hypsometry, aspect, debris cover, and the presence of proglacial lakes and ice cliffs, for example. Combined with uncertainties in ice thickness and future climate scenarios, the timing of peak water runoff and the rate of its subsequent decline remain key unknows (Maurer et al., 2019; Nie et al.,2021; Su et al., 2016; Zhao et al., 2019). It is therefore critical to monitor and analyze glacier change to improve our understanding of its climate drivers, and to assess its impacts on glacier-fed river basins.

Compared with the interpolation of sparse in-situ measurements, satellites can observe glacier change over much larger areas of remote terrain (Wang et al., 2021a). In recent years, our understanding of the state of TP glaciers has been greatly improved by the increasing coverage and accuracy of multi-source remote sensing observations of glacier area, volume and mass change from KH-9 (Hexagon military satellites), Landsat, ASTER, ICESat altimetry, and other Digital Elevation Models (DEMs) constructed by geodetic techniques, and from GRACE gravimetry ( Guo et al., 2015; Kääb et al., 2012; Wang et al., 2021a; Zhou et al., 2018). Based on the KH-9 images and SRTM, for example, Zhou et al. (2018) found that from the mid-1970s to 2000 glaciers in the northwest TP thinned at $-0.11 \pm 0.13$ m w.e. $a^{-1}$ to $0.02 \pm 0.10$ m w.e. $a^{-1}$ while those in the southeast part thinned faster at $-0.30 \pm 0.12$m w.e. $a^{-1}$ to $-0.11 \pm 0.14$m w.e. $a^{-1}$. Brun et al. (2018a) employed ASTER DEMs from 2000-2006 and showed that glacier mass balance in High Mountain Asia varied from $-0.62 \pm 0.23$ m w.e. $a^{-1}$ in eastern Nyainqentanglha to $+0.14 \pm 0.08$ m w.e. $a^{-1}$ in the Kunlun Mountains, and averaged $-0.14 \pm 0.14$ m w.e. $a^{-1}$ over the large Inner TP that includes WNR. Maurer et al. (2019) found a doubling of the Himalayan average loss rate between the periods 1976-2000 $(-0.22 \pm 0.13$ m w.e. $a^{-1})$ and during 2000–2016 $(-0.43 \pm 0.14$ m w.e. $a^{-1})$ using KH-9 and ASTER DEMs. These studies showed that glacier changes on and around the TP have marked spatial and temporal heterogeneity, likely associated in part with variable glacial sensitivity to climate change (Yao et al., 2012).

The drivers of regional glacier loss include, for example, a jump in mean annual temperature and precipitation in the Yarlung-Zangpo River basin around 1997 (Wang et al., 2021b) and an accelerating warming trend over the TP between the periods 1980–1997 and 1998–2013 (from 0.21 °C to 0.25 °C decade$^{-1}$) (Duan & Xiao, 2015).Modulating the effect of these climatic drivers are local factors including glacier topography, debris-cover and, glacial lakes (Brun et al., 2018b, 2019; Ke et al., 2020; Maurer et al., 2019; Pandey et al., 2017; Yao et al., 2012). Some studies have analyzed the melt-inhibiting effect of debris cover and melt-promoting effect of proglacial lakes on glacier ablation since 2000 (Ke et al., 2020; Vincent et al., 2016), but with the potential for KH-9 in 1976, SRTM in 2000, ASTER in 2000-2020 and aerial mapping Landsat 1976-2020 through time, we are now able to assess glacier area and mass change in the WNR in relation to both regional climatic derivers and local modulating factors.

The WNR, in the south-eastern TP (Figure 1), is located in the transition zone between the two large-scale atmospheric circulation patterns characterized respectively by dominant westerlies and the Indian summer monsoon. It holds an abundance of glaciers and glacial-fed lakes, notably Nam Co Lake (Figure 1), whose rising water levels indicate a water imbalance

primarily due to recently intensified glacier melting (Bolch et al., 2010), as supported by mass balance data from Zhadang
Glacier and other hydrological observations from 2007 to 2011 (Zhou et al., 2013). The number and area of supraglacial lakes
(of >0.0036 km$^2$) in the WNR also increased between 1976 and 2018 by 56% and 35% respectively due to the increase in
glacial meltwater (Luo et al., 2020). In the relatively densely-populated Lhasa Basin to the southeast of WNR, Lin et al. (2020)
found that a water imbalance also existed using the first and second Chinese Glacier Inventory in 1960 and 2009. Despite these
extensive changes and large affected population, logistical constraints have meant that in situ glacier mass balance records are
limited to a few low-lying, small glaciers that are unlikely to be representative of the broader region (Kääb et al., 2012; Li &
Lin, 2017; Yao et al., 2012).Similarly, glacier volumes in the Chinese Glacier Inventory were primarily calculated indirectly
by area-volume scaling, and limited direct observations mean that these volume have larger uncertainty (Bahr et al., 1997;
Bahr et al., 2015). Detailed investigation of the WNR glacier area change and mass balance on a longer time scale is therefore
a high priority.

Investigations of WNR glacier area have so far focused on the period before 2014 ( Bolch et al., 2010; Wu et al., 2016).

For glacier mass balance, most studies focus on after 2000 (Li & Lin, 2017; Neckel et al., 2014; Ren et al., 2020; Zhang &
Zhang, 2017). There are limited discussions on local glacier changes in the WNR region from before 2000, although Zhou et
al. (2018) included this area in his study of glacier mass balance on the TP and its surroundings from the mid-1970s to 2000
and did not present the characteristics of glacier changes in detail. Furthermore, the warming rate of the TP is heterogeneous
both spatially and temporally in recent decades (Duan & Xiao, 2015; Wu et al., 2015). Under a changing runoff regime (Lin
et al., 2020), the lack of a detailed survey of glacier changes over a long time scale is a major impediment to water resource
management and decision-making (Lutz et al., 2014).

The key purpose of this study is therefore to provide an internally consistent dataset of glacier area and mass change in

the WNR over the past 44 years, and comparative analysis of the impact of topography, debris and proglacial lakes on glacier
change during 1976 - 2000 and during 2000 - 2020. Although the area of both debris-cover and lake terminating glaciers are
relatively small, the characteristics of their influence on individual glaciers can be used as a reference for glacier changes in
other regions. We have compiled a complete glacier inventory in the years 1976, 2000, 2014 and 2020 with the Landsat and
KH-9 images and quantified the geodetic glacier mass balance during 1976–2000 and 2000–2020 with DEMs derived from
KH-9, ASTER and SRTM3.0. We report area and mass changes for periods 1976 to 2000 and from 2000 to 2020, and we
examine and compare the influence of topographic, climatic, and glaciological factors on these changes during 1976-2000 and

2000-2020.

**2. Materials and methods**
**2.1 Study area**

The WNR has a mean slope of 15° and its elevation spanning 4150-7125 m, with an average of 4930 m in the whole

region. Its primary mountain ridge runs 230 km in a southwest−northeast direction, bounded by the Nam Co basin to the north
and the Lhasa River basin to the south (Yao et al., 2010). Nam Co Lake, the second largest after Selin Co Lake on the TP, is
mainly fed by glacier meltwater (Luo et al., 2020; Zhang et al., 2017). The Lhasa River basin to the southeast is a major branch
of the Yarlung Zangbo River and forms part of the route taken by the warm and humid monsoon airflow into the plateau,
making it warmer and wetter than the Nam Co basin (Luo et al., 2020). The annual air temperature and precipitation in the
WNR range from −0.6℃ to 2.8℃ and 37 mm to 500 mm, respectively (Yu et al., 2013).
Being in a climatic transition zone, the glaciers in this area range from the maritime-influenced glaciers of Gangrigabu
(southeast TP) to the subcontinental and continental-type glaciers of the Tanggula mountains (Li & Lin, 2017). There are 845
glaciers covering 675.85 km$^2$ and 15 debris-covered glaciers with a total area of 71.74km$^2$ (10.61% of the total glacier area in
the WNR ) (RGI 6.0) (RGI Consortium, 2017). Of these, only the small, ~3km$^2$ Zhadang and Gurenhekou glaciers (red polygon
in Figure 1) have in-situ observations available to validate the satellite observations, and these run from 2005 to 2010 (Yao et
al., 2012).
**2.2 Methods and data**
**2.2.1 Glacier outlines**
We identified glacier boundaries mainly from Landsat MSS/ETM+/OLI scenes from various years (Table S1),
orthorectified automatically by the USGS using the level 1T SRTM3 DEM (from http://glovis.usgs.gov/). We selected high-
quality images with minimal cloud and snow coverage between June and November and used a semi-automated approach with
a TM3/TM5 band ratio (2.0 ± 0.2) to produce glacier outlines. This method is widely used and appropriate for glacier mapping
over large study areas (Guo et al., 2015; Ye et al., 2017). We used a 3 × 3 median filter to eliminate isolated pixels and likely
to have been misclassified due to debris or boulders on the glacier (Bolch et al., 2010). We manually checked and edited the
glacier outlines, including the debris-covered glaciers, with height change maps and a coherence map formed by Sentinel-1
images observed on 2016-08-05 and 2016-08-29, to help distinguish debris-covered ice from ice-free areas. Finally, referring
to the second glacier inventory, we assigned contiguous ice masses to drainage basins in order to obtain a glacier inventory.
**2.2.2 Glacier elevation change**
We used the KH-9 images and SRTM DEM (version 3) to estimate glacial elevation changes for the period 1976 to 2000,
and ASTER DEMs for the period 2000-2020.
2.2.2.1 DEM data
The declassified KH-9 images were obtained by the Hexagon mission from 1971 to 1986, with a ground resolution of 6
to 9 m (Surazakov & Aizen, 2010). We downloaded images from 1976-01-07 via the Earth Explorer user interface
(https://earthexplorer.usgs.gov) and adopted the Hexagon Imagery Automated Pipeline method to generate digital elevation
models. This method is coded in MATLAB and uses the OpenCV library for Oriented FAST and Rotated BRIEF (ORB) feature
matching, uncalibrated stereo rectification, and semiglobal block matching algorithms (Maurer & Rupper, 2015).
The SRTM mission carried out in February 2000 produced two types of DEM datasets, the C-band DEM with a coverage
range of 60°N ~ 60°S and the X- band DEM with a smaller coverage. We used version 3 of the C-band SRTM DEM
(https://earthexplorer.usgs.gov/) at 1-arc-second resolution (about 30 m) in our primary processing and masked out areas with
gaps in the unfilled SRTM3 version 2.1 DEM at 3-arcsencond resolution (about 90 m).
The ASTER instrument was launched on the Terra satellite in December 1999 and a single DEM covers approximately
3600 km². We downloaded 250 'Data1. l3a.demzs' ASTER DEMs at 30 m resolution in geotiff format with cloud coverage of
less than 40% from the METI AIST Data Archive System (MADAS) satellite data retrieval system (https://gbank.gsj.jp/madas).
After cloud and outlier removal we fitted a linear regression through the time series of co-registered ASTER DEMs and set
the minimum stack interval per pixel to 15 years to estimate the rate of elevation change for each 30-m pixel (Maurer et al.,

2019).

2.2.2.2. Co-registration and bias correction of DEMs
All DEMs were co-registered to the SRTM master DEM using a standard elevation–aspect optimization procedure (Nuth
& Kääb, 2011). Then, the elevation correlation deviation of all the DEMs was corrected by a third-order polynomial. In addition,
we used a 2km buffer zone around the union of glacier boundaries to define stable (unchanging) terrain for DEM alignment,
bias correction, and uncertainty calculation. Figure. S1a shows the coverage of the KH-9 images and the number of valid
ASTER DEMs grids after removal of clouds and outliers in the buffer. The glacier area covered by the dataset from 1976 to
2000 and from 2000 to 2020 accounted for 70.85% and 81.94% of the total glacier area, respectively (shown in Figure. S1b,
c). We used only the area common to both of these datasets to measure elevation change between the 1976-2000 and 2000-
2020 periods. After correction for alignment and elevation-related deviation, apparent elevation changes over stable terrain
(masked glaciers and lakes in square buffer zone) had no change trend with elevation, slope and aspect, as shown in Figure.
S2.
2.2.2.3. SRTM Penetration depth correction
Over the WNR, the average penetration depth of C-band SRTM is 1.67 ± 0.53 m calculated using X- band SRTM DEM
as the reference (Li & Lin, 2017). Linear regression between the glacier elevation and penetration showed that the penetration
depth varies from 1.29 m to 2 m at altitudes of 5550 m and 6250m respectively (Li & Lin, 2017). We used this more accurate
linear altitude-dependent correction, and the result is similar to several other study regions on the TP (Gardelle et al., 2013;
Kääb et al., 2012; Li & Lin, 2017).
2.2.2.4 Glacier mass change
Estimation of average glacier thickness changes based on elevation difference maps involves noise filtering and glacier-
hypsometry-weighted averages in an approach widely employed in to calculate regional glacier mass balance where glacier
thinning is highly dependent on altitude. Firstly, we subjected the thickness change maps to outlier removal using a 5 m a$^{-1}$
threshold. We then masked slopes > 45°, where uncertainties are large, before visually inspecting the final thickness change
maps. We additionally masked out any remaining anomalous pixels, which occurred almost exclusively in low-contrast, snow-
covered accumulation zones. Finally, we separated thickness changes into 50-m elevation bins by referring to the SRTM at
different spatial scales, i.e., the whole glacierized area, sub-regions, different glacier types and individual glaciers of area >2
km$^2$. In each altitude bin, we filtered out any height-change values that differed by more than three standard deviations from
the median and removed any bins with less than 100 pixels. For elevation bins with no observations (mostly over the low- and
high- elevation limits), we assumed zero mean elevation changes. We calculated the mean glacier thickness changes for the
spatial unit/group (dh) as a hypsometric average:
$$dh = \sum_{i=1}^{n} \frac{S_i}{S} \cdot \overline{dh_i} \qquad (1)$$

where $i$ and $n$ denote the $i^{th}$ 50-m elevation bin and the number of total bins respectively, $S_i$ is the glacier area of the $i^{th}$ elevation
bin, S is the total glacier area, and $\overline{dh_i}$ is the mean $dh$ in the bin.
We calculated the final geodetic mass balance ($B$) using equation (2).
$$B = dh_i \times \frac{\rho_{ice}}{\rho_{water}} \qquad (2)$$

We translated glacier thickness changes into mass balance by the ratio of column-averaged glacier density, $\rho_{ice}$ (850 kg m$^{-3}$) to
water density ($\rho_{water}$, 1000 kg m$^{-3}$).
**2.2.3 Uncertainty**
2.2.3.1 Uncertainty of glacier area
Similar to previous studies (Wu et al., 2016; Ye et al., 2017), we obtained the uncertainty of glacier area ($\delta_s$) using equation

(3).

$$\delta_s = L_c E_{pc} + L_d E_{pd} \qquad (3)$$

Where $L_c$ and $L_d$ represent the lengths of the clean-ice and debris-covered glacier outlines, and $E_{pc}$ and $E_{pd}$ denote the positional
accuracies. We calculated the uncertainty in glacier area change ($\delta_{sc}$) by combining the area uncertainties using equation (4).
$$\delta_{sc} = \sqrt{(\delta_{s1})^2 + (\delta_{s2})^2} \qquad (4)$$

Guo et al. (2015) compared glacier outlines derived from Landsat-images with real-time kinematic differential GPS (RTK-
DGPS) measurements and found an average difference of ±11 m and ±30 m for the delineation of clean and debris-covered
ice. Using a buffer size of 10 m for areas from the Hexagon images (Bolch et al., 2010), our combined uncertainty in glacier
area is 3.9%, 5.1%, 5.1% and 5.9% in 1976, 2000, 2014, and 2020, respectively.
2.2.3.2 Uncertainty of glacier thickness change

The uncertainty in surface-elevation change derived from ASTER DEMs can be estimated using the point elevation error

($E_{pt}$) and extrapolation error ($E_{ext}$) (Nuth & Kääb, 2011; Maurer et al., 2016).
$$\delta_{hi} = \sqrt{(\frac{E_{pt}}{\sqrt{n_i}})^2 + (\frac{E_{ext}}{\sqrt{n_i}})^2} \qquad (5)$$

$$n_i = \frac{n_{ib} * r^2}{\pi * d^2} \qquad (6)$$

$$\delta_h = \sqrt{\sum_{i=1}^{i=n}(\delta_{hi} * \frac{S_i}{S})^2} \qquad (7)$$

Here, $E_{pt}$ refers to the standard deviations of the relative elevation change over the off- glacier areas, $E_{ext}$ is the standard
deviations of glacial elevation change within each 50-m bin, $n_{ib}$ and $n_i$ represent the total number of pixels and the number of
independent measurements of pixels respectively, $r$ is the DEM spatial resolution (30 m in our study), and $d$ is the
autocorrelation length. We used an autocorrelation length of 500 m was employed, which is a conservative value based on
semivariogram analysis of mountainous regions in previous studies (Brun et al., 2018a; Maurer et al., 2019). We combined,
the uncertainty of surface-elevation changes derived from the KH-9 DEM with the SRTM penetration uncertainty, estimated
as ±0.53 m (Li & Lin, 2017). This study ignored the errors caused by seasonal changes in glacier thickness due to the lack of
observations of such seasonal changes.

We estimated the overall uncertainty in the total glacier mass change ($\delta m$, in kg) by including the uncertainty in the

assumed ice/firn/snow density ($\delta \rho = 60$ kg m$^{-3}$, which is 7.1% of $\rho_{ice} = 850$ kg m$^{-3}$), errors in glacier area ($\delta_S$, m$^2$) and glacier
elevation change ($\delta_h$, m), using equation (8).
$$\delta_m = \sqrt{(S \cdot dh \cdot \delta_\rho)^2 + (\delta_s \cdot dh \cdot \rho_{ice})^2 + (S \cdot \delta_h \cdot \rho_{ice})^2} \quad (8)$$

**2.2.4. Lake data**

We identified glacier-marginal lakes as those lying within 50 m of a glacier boundary, using lake data for the 1970s-2018

(Luo et al., 2020; http://data.tpdc.ac.cn).
**2.2.5. Meteorological data**

There are three meteorological stations adjacent to the WNR, at Bange (31°23′N, 90°01′E, elevation of 4700 m), Lhasa

(29°40′N, 91°08′E, elevation of 3648 m), and Damxung (30°29′N, 91°06′E, elevation of 4200 m). We obtained air temperature
and precipitation data during 1976-2020 from the Climatic Data Center, National Meteorological Information Center, of the
China Meteorological Administration.

We also obtained gridded data of precipitation and temperature with spatial resolution of 0.1° × 0.1° and 3-h time interval

for 1979-2018 from the China Meteorological Forcing Data (Ding et al., 2020; http://data.tpdc.ac.cn), which has been widely
utilized in land-process, hydrological modelling, and other studies (Qiao et al., 2021; Wang et al., 2021b). This dataset is made
by fusing the conventional meteorological observation of China Meteorological Administration based on the Princeton
reanalysis data, GLDAS data, GEWEX-SRB radiation data, and TRMM precipitation data as the background field (He et al.,
2020; Yang et al., 2010).
**2.2.6 Hydrological Data**
In order to assess hydrological changes under glacier retreat, we have collected runoff data of the Lhasa River station
during 1976-2013 and the Yangbajain station during 1979-2013 from the Tibet Autonomous Region Hydrology and Water
Resources Survey Bureau.
We calculated the ratio of total glacier mass change to runoff in Lhasa River basin ($R_r$, %) and the total lake water
storage change of Nam Co Lake ($R_l$, %) as follows:
$$R_r = \frac{\Delta M * S_g}{S_r R_a} \qquad (9)$$

$$R_l = \frac{\Delta M * S_g}{\Delta V} \qquad (10)$$

Where $\Delta M$, $S_g$, $S_r$, $R_a$, $\Delta V$ represent average annual glacier mass balance, glacier area, area of the Lhasa River basin,
average runoff depth, lake water storage increase.
**3.Results**
**3.1 Glacier area change**
There were 921 glaciers with a total area of $589.17 \pm 31.72$ km$^2$ in 2020 in the WNR (Figure 2a). Small glaciers dominated
the number (those ≤1 km$^2$ occupy 83.17% of the total number) and a large proportion of the area (those ≤1 km$^2$ occupy 30.42%
of the total area). Glaciers larger than 5 km$^2$ accounted for 21.39% of the total area and only 1.63% of the total number. Glaciers
were mainly distributed in the eastern-oriented zone with an altitude of 5600-6100m and a slope of 5-40° (Figure 2b, c, d).
Glaciers in the WNR experienced significant retreat from 1976 to 2020 and altitude, slope and aspect all appear to have
influenced this retreat (Figures 3 and 4). The glacier area decreased from $884.90 \pm 29.71$ km$^2$ in 1976 to $589.17 \pm 31.72$ km$^2$
in 2020, with an average annual decrease of $-0.76 \pm 0.11$ % a$^{-1}$. The retreat rate of glacier area in 2000-2020 (-1.17% a$^{-1}$) was
more than twice as fast as in 1976-2000 ($-0.54 \pm 0.21$ % a$^{-1}$) (Figures 3 and 4, Table 1). The glacier area declined faster in the
northeast and southwest but slower in the middle, except for a few small glaciers with an area of less than 2 km$^2$ during 1976-
2000. During 2000-2020 the glacier area receded faster in the whole region except for a few small glaciers with an area of less
than 1 km$^2$ (Figure 3). Retreat was greatest in the area classes of 1-3 km$^2$ and 3-5 km$^2$, and glaciers with significant areal retreat
were mainly distributed below 6,000 m altitude. Glaciers in the Nam Co basin retreated slightly faster than those outside this
basin between 2000 and 2020. Retreat was particularly rapid at lower altitudes and decreased at higher elevations. As for the
effect of slope and aspect, glacier retreated more rapidly with increasing slope between 5° and 40°, but the retreat rate decreased
as slope increased between 0°-5° and 40°-60°, where relative few glaciers are distributed. In each elevation band, we found
a positive correlation between areal retreat rates and slope (faster retreat with steeper slope) for most elevation bands
and in both time periods (Figure 9 a and b). The only areas where this relationship differed were on flat or shallow slopes
at lower altitudes (slopes below about 5° at elevations below about 5500 m, e.g., blue lines in Figure 9a) which also
experienced relatively rapid retreat. During both 1976-2000 and 2000-2020, the retreat rate was smallest on the north-facing
slopes. During 1976-2000, retreat was most rapid in the southeast quadrant, while from 2000 to 2020, rapid retreat occurred
at similar rates in all aspects other than north and southeast, i.e., the effect of aspect on glacier area retreat varied in space and
time.
Besides, we noticed that the area decreased but glacier number increased in WNR (Figure 2a). The reason for this is that
intact glaciers break down into several smaller glaciers in the process of glacier ablation, e g., a large glacier in 1976 may
become several smaller glaciers in 2020 (shown in Figure S3).

**3.2 Geodetic mass balance**

Glacier height changes for the past 44 years, are shown in Figure 5. Substantial and near-ubiquitous thinning occurred in
the WNR during 1976-2020 (with mean surface lowering of -0.37 ±0.13 m $a^{-1}$, a water-equivalent loss rate of -0.31 ± 0.12
m w.e. $a^{-1}$ or a mass loss rate of -0.26 ± 0.09 Gt $a^{-1}$), with a widespread increase in the most recent decades. From 1976 to
2000, glaciers experienced a mean thinning rate of -0.31 ± 0.10 m $a^{-1}$ (-0.26 ± 0.09 m w.e. $a^{-1}$) or -0.24 ± 0.08 Gt $a^{-1}$. From
2000 to 2020, the mean elevation rate was -0.44 ± 0.13 m $a^{-1}$, (0.37 ± 0.12 m w.e. $a^{-1}$) or -0.29 ± 0.09 Gt $a^{-1}$. Several glacier
tongues have suffered severe thinning, exceeding -1.5 m $a^{-1}$ from 1976 to 2000, notably several long, debris-free glaciers on
the south-western slope. From 2000 to 2020, the range of glacier tongues with losses exceeding 1.5 m $a^{-1}$ expanded, and losses
were greater in the central WNR (see, the red rectangular box in Figure 5). In both 1976-2000 and 2000-2020, the glacier
thinning rate was slightly higher inside the Nam Co drainage basin than outside it (Table 2, Figure. 6), though these rates do
not differ by more than their combined uncertainties.
For glaciers with an area of more than 2 km², we found high loss rates in the northeast, followed by the southwest, and
moderate in the middle during 2000-2020, but there no obvious spatial varied trend of mass loss during 1976-2000 (Figure 7).
Mass loss was substantially more intense in 2000-2020 with no glaciers in a state of positive balance (Figure 7, blue dots) and
loss from some glaciers in the northeast exceeded -0.6 m w.e. $a^{-1}$. Moreover, we found that glacier area retreat and mass loss
was not synchronized between the two periods 1976-2000 and 2000-2020. The glacier with the fastest area retreat did not
correspond to the glacier with the fastest mass decrease, and the spatial varied trend of glacier area retreat rate was inconsistent
with that of mass loss rate (Figure 3 and 7).
Finally, glacier elevation change as a function of elevation slope and aspect are shown in Figure 8. Elevation is inversely
correlated with thickness change, while slope and aspect appear to have a weak relationship with thickness change. In both
1976-2000 and 2000-2020, the elevation change rate was the largest at lower altitudes, and gradually decreased with the
increasing of altitude. The thinning rate also exhibited a weak inverse relationship with slope, becoming somewhat stronger in
the 2000-2020 period. A very similar but inverse to relationship between slope and glacier area change rate in each
elevation band is that thinning rates were highest on shallow slopes and decreased over steeper slopes, except for flat or
shallow slopes at lower altitudes where thinning rates were relatively low (Figure 9 c and d). For the impact of aspect,
thinning for 1976-2000 was most rapid in the south-west and north-west quadrants, but by 2000-2016 high thinning rates were
affecting all aspects, i.e., the effect of aspect on thinning rates also varied through time.
**3.3 The effect of debris-cover and proglacial lakes on glacier mass changes**
In our WNR study area, there are five debris-covered glaciers, covering $55.42 \pm 1.25$ km$^2$ in 1976 and $51.59 \pm 1.77$ km$^2$
in 2000. Lake-terminating glaciers occupied a similar proportion, with area of $70.29 \pm 2.69$ km$^2$ in 1976, and $49.60 \pm 1.82$ km$^2$
in 2000. Only one glacier was both covered by debris and terminated in a pro-glacial lake.
The thinning rate of different types of glaciers varied somewhat, though with greater uncertainty given the relatively small
sample. (Figure 10, Table 3). During 1976-2000, the lake-terminating glaciers thinned more rapidly, followed by the regular
and debris-covered glacier types. From 2000 to 2020, the ablation rate of debris-covered glaciers was slightly lower than that
of regular glaciers at low altitudes, but progressively greater at higher altitudes, leading to a slightly more negative total mass
balance for the debris-covered type. In the same period, the thinning of lake-terminating glaciers continued to exceed that of
regular glacier. Our results suggest that debris cover in the WNR suppressed glacier thinning to some extent and enabled the
debris-covered ice to survive at lower elevations than adjacent clean ice glaciers. In contrast, a glacial lake at the end of a
glacier accelerated its retreat, and this behavior was more pronounced at lower elevations.
**4.Discussion**
**4.1 Comparison to previous studies**
**4.1.1 Glacier area change**
Based on space-borne imagery, we found that glacier area in the west Nyainqentanglha Range (WNR) has changed by -
12.98±4.91 during 1976-2000 and -23.45±5.99% during 2000-2020. The comparison between this, Chinese Glacier Inventory
(CGI) I and CGI II over WNR, and previous studies and is shown in Tables S2 and S3. The CGI II of WNR in 2009 are in
good agreement with the areal retreat trend in our study (also shown in Figure 2). The CGI I of WNR in 1970 is slightly smaller
than the glacier area in 1976 in our study, but it is within the margin of uncertainty. The CGI I was mapped based on the
Chinese topographic maps, while glacier area in our study was mapped based on Landsat Images. The difference between them

might come from this difference in data source used to extract the glaciers outlines. Besides, Frauenfelder & Kääb, (2009) reported that there are georeferencing errors in the areas in GGI I. Differences between studies may have arisen from the georeferencing errors in the areas for 1970 used by Shangguan et al. (2008) and Wu & Zhu (2008) which came from the CGI I. Discrepancies may also have arisen from differences in the methods used to distinguish glaciers from seasonal snow, and debris-cover glaciers from neighboring moraine or rock slopes (Bolch et al., 2010). The deviation between our results and those from Wu et al. (2016) and Wang et al. (2012) over the whole WNT and the southwest WNT is within the margin of uncertainties. In addition, the 789.15 km$^2$ area reported for the WNR by RGI V4.0 which used Landsat images obtained on 2001-12-06 agrees with our result.

**4.1.2 Glacier mass balance**

Field measurements of mass balance are available from small Zhadang glacier for 2005-2008, and Gurenhekou glacier for 2005-2010 on the southeastern slope of the WNR (Table S4). Although the period of our study is longer and provides a much larger sample size, the mass balance results are similar to these field measurements.

Previous studies have also reported region-averaged glacier mass balance over a similar spatial extent to ours, obtained from DEMs using various sensors (Table S5). Our results during 2000-2020 are more negative than those of Neckel et al. (2014), Li & Lin (2017) and Zhang & Zhang (2017), but agree within the uncertainties over comparable time periods, even though these studies differ in data processing, glacier mask, penetration correction and data coverage. For comparison, we calculated the change for the 2000-2014 period from ASTER DEMs (Figure 11). Our estimated mass balance in this area (-0.28±0.15 m w.e. a$^{-1}$) is very similar to the other studies (Table S5). It is also similar to that of 1976-2000, suggesting that the more strongly negative average for the longer 2000 to 2020 period (-0.37±0.12 m w.e. a$^{-1}$) is mainly due to the intensified glacier ablation after 2014, although cloud-free ASTER data are insufficient for direct calculation of the mass balance from 2014-2000. The glacier area retreat during 2014-2020 (1.53±1.14 % a$^{-1}$) is also faster than that during 2000-2014 (1.13±0.42 % a$^{-1}$), though the change is within the uncertainties. This interpretation is supported by Ren et al. (2020) who also calculated a higher 2013-2020 thinning rate (-0.43±0.06 m w.e. a$^{-1}$) twice as negative as in 2000-2013. Though the difference in rate is within the combined uncertainties for these periods, this apparent acceleration in thinning in WNR (from -0.26 ± 0.06 m w.e. a$^{-1}$ in 1976-2000 to -0.37±0.15 m w.e. a$^{-1}$ in 2000-2020), is similar to the broader regional pattern of accelerating loss across the Himalayas and Kangri Karpo Mountains (Maurer et al., 2019; Wu et al., 2018, 2019).

**4.2 The influences of debris-cover and proglacial lakes on glacier mass changes**

Debris can inhibit or enhance glacial ablation depending on its thickness (Maurer et al., 2016). A shallow layer of debris usually enhance melt rates due to its low surface albedo, while thicker layers could suppress melt rates through thermal insulation (Reid et al., 2012). Our results (Table 3) suggest that the debris-covered glaciers in our study thinned more slowly

than the regular, debris-free glaciers in the 1976-2000 period, though the difference is not statistically significant and the small sample size (5) of the debris-covered glaciers compared to regular glaciers (>600) limits our ability to compare these classes. In the 2000-2020 period, the thinning rate of the debris-covered glaciers increased significantly, to double its previous rate, though it remains indistinguishable from the thinning rate for regular glaciers at that time. While several previous studies indicated that on the glacier-scale, debris-covered glaciers thin more slowly than debris-free glaciers(Nicholson & Benn, 2006; Scherler et al., 2011; Vincent et al., 2016), large-scale geodetic studies reported no significant differences in the thinning rates between debris-covered and clean glaciers on time scales more than a decade after 2000 (Brun et al., 2019; Ke et al., 2020; Maurer et al., 2019), a finding that is supported by this study. Banerjee (2017) suggested that the thinning rate of a debris-covered glacier is initially slower than that of a similar clean glacier at the early stage of warming but subsequently matches and then overtakes the clean counterpart. In this theory, the time required for their respective melting rates to cross is controlled by the rate of warming, with little difference between their thinning rates at low rates of warming (Banerjee, 2017). The large difference in the 1976-2000 mean melt rates of the regular versus debris-covered glaciers in our study provides some supports for this theory, but a larger sample with lower uncertainty is needed to verify this.

For debris-covered glaciers, the area of debris cover actually increased from $6.60\pm1.15\,\text{km}^2$ in 1976 to $7.37\pm1.48\,\text{km}^2$ in 2020 in our study (Table S6), and we note that this is not necessarily inconsistent with an overall glacier retreat. This is because increased melt rates that lead to surface lowering drive retreat of the glacier front, while also promoting a greater concentration of debris on the wider surface of glacier ablation area as more debris melts out from ice below. A spatial expansion of the debris layer has, for example, been observed on different debris-covered glaciers during retreat and sustained mass loss. (Kirkbride & Deline, 2013; Stokes et al., 2007; Tielidze et al., 2020; Xie et al., 2020). Unfortunately, no data are available relating to changes of the thickness of the debris cover itself, and we assume that all glacier thickness changes resulted from loss of ice, without considering the thickness change of the debris cover layer. We think that this is reasonable in because in most area, debris layers are typically thin (order of 1 meter or less) and compared to elevation changes we map, and because most debris cover in the ablatio area emerge from englacial transport rather than direct deposition by new, local rock fall(e.g., McCarthy et al. 2017), so changes in the debris-layer thickness represent a redistribution of existing glacier volume, not a change in volume.

Glaciers with proglacial lakes can experience relatively high mass loss through calving and thermal undercutting (Maurer et al., 2016; Thompson et al., 2012) and the expansion of such lakes can cause dynamic thinning to propagate up-glacier (Ke et al., 2020). Glaciers terminating in proglacial lakes in our study area had the highest mean thinning rates of all the classes in both time periods, and more negative mass balance compared to both regular and debris covered glacier during 2000-2020.

The area of debris-covered glaciers and lake-terminating glaciers decreased, while surface lowering also accelerated, mainly driven by the continuous increase in temperature in the WNR region during 1976-2000, especially after 2014 (Figure 12 and Figure 13), which was discussed in 4.3. In terms of the number and area of lake-termination, we identified glacier-

marginal lakes as those lying within 50 m of a glacier boundary. As glaciers retreat, the distance between the end of the glacier
and their proglacial lake increased, and some of lake-terminating glaciers in 1976 no longer belonged to lake-terminating class
in 2000. This helps for explain the area decreased for this glacier class in Table 3.

**4.3 Topographic and climatic controls of varying glacier mass loss**

If climate is the driving force behind glacier change, topographical parameters can modulate this change (Pandey et al.,
2017). Controls on glacier thickness and areal change are complicated, however, with additional factors including local
variations in climate, glacier thickness, morphology, the presence of proglacial/supraglacial lakes and debris cover, and latitude
and longitude (Brun et al., 2018b,2019; Ke et al.,2020; Maurer et al., 2019).
We found that both glacier areal retreat rate (Figure 4b) and thinning rate (Figures 8 and 9) generally decreased with
increasing altitude, agreeing with previous studies (Li & Lin, 2017; Wu et al., 2016; Ye et al., 2017; Zhou et al.,2019). However,
the effect of slope and aspect on glacier thickness has been rarely studied. We found that in the slope range of 8-40°, where
the glaciers were mainly distributed, the rate of areal retreat increased as slope increased (Figure 4c), but the thinning rate
decreased (Figure 8b). This occurred because steep slopes are associated with thinner ice (Linsbauer et al., 2012), which
means that any given thinning rate will tend to drive more rapid areal retreat on steeper slopes as the thinner ice there is
depleted first, explaining the broadly positive correlation between retreat and slope. Besides, steeper slopes are biased
towards higher elevations, where the colder climate leads to slower thinning rates (dh), explaining the broadly negative
correlation between slope and thinning rate. The somewhat different behavior of the low-elevation flat areas (relatively
rapid retreat, relatively slow thinning, Figure 9) may in part reflect the modulating effects of proglacial lakes (quicker
retreat) and thicker debris cover (slower thinning) near the terminus. Overall, the relationship between aspect and both
areal retreat and thinning was spatially inconsistent and varied in time (Figures 4d and 8c).
Mean glacier mass thinning and retreat rates were consistently higher in the Nam Co basin than Lhasa River basin (Table
1 and 2), in agreement with Bolch et al.(2010) and Li & Lin (2017), and the glaciers in central WNR showed particularly
strong melting from 2000 to 2020. While the glacier distribution on the TP broadly follows the regional atmospheric circulation
pattern (Yao et al., 2012), the variability in glacier loss within regions cannot always be fully explained by the changes in
precipitation and temperature on this scale (Wu et al., 2018).
The increasingly-negative mass balance through time is consistent with the temperature record from the three weather
stations that shows a consistent warming trend (averaging 0.0485 °C a$^{-1}$) (Figure 12) and gridded temperature data showing a
more rapid increase during 2000-2018 than 1979-2000 (Figure 13), alongside precipitation that increased during 1979-2020
but decreased during 2000-2018. The accelerated warming from 2014 to 2018 (red rectangle in Figure 13 (2014-2018))
corresponds geographically to the substantial central-WNR glacier thinning highlighted in the red rectangle in Figure 5.
Precipitation also increased substantially in this region from 2014 to 2018, and glacier melting can be particularly intense
under combined warm and wet conditions (Li et al., 2020; Oerlemans & Fortuin, 1992).
While the overall, the trends of temperature and precipitation in the ablation season (June to September) and
accumulation season (October to December and January to May) were similar to annual changes, the temperature and
precipitation data from 2014 to 2018 described above offer a compelling explanation for the main temporal and spatial
variations in glacier change in the WNR, particularly the high rates of thinning from 2014-2018. They do not directly explain
why the Nam Co glaciers thinned more rapidly than elsewhere, however. Other possible explanations include difference in the
impact of black carbon and dust in reducing surface albedo (Lau et al., 2010; Li & Lin, 2017; Ming et al., 2008), and Qu et al.
(2014) did observe a decrease in albedo at Zhadang glacier (Nam Co basin) from 2001-2012.
**4.4 Hydrological effect**
The glacier melt contribution to streamflow decreases significantly from the glacier terminus to the lowlands as it
becomes diluted by other water sources (Kaser et al.,2010; Lutz et al., 2014; Pritchard, 2019) and this is reflected in our
finding that the average annual glacier mass loss during 1976-2014 ($-0.26\pm0.14$ m w.e.a$^{-1}$) equates to $8.5\pm4.6\%$ of the mean
annual runoff depth for the Yangbajain basin, in the upper reaches of Lhasa River (location shown in Figure 1(a)), but only
$1.6\pm1.0\%$ for the Lhasa Riber basin as a whole.
Through this period, the annual runoff in the Yangbajain basin showed a significant increase trend of 1.32 mm a$^{-1}$ and
the Lhasa River basin a non-significant increase trend of 0.84 mm a$^{-1}$ (Figure S4). Increasing runoff may in part be explained
by a coincident 1.36 mm a$^{-1}$ increase in precipitation observed over the Lhasa River basin (Figure 12(b)), though the glacier
ablation increase in Lhasa River basin and Yangbajain basin ($4.63\pm2.49$ mm a$^{-1}$ and $23.52\pm12.67$ mm a$^{-1}$ respectively) were
substantially greater than the increase in precipitation, and evaporation losses from glacier melt water tend to be substantially
smaller than those from evaporation of precipitation over the basin (Pritchard, 2019), suggesting that increased glacial
meltwater primarily drove increased runoff. This is supported by Lin et al. (2020) who attributed increase streamflow at
Yangbajain Station to accelerated glacier retreat, and Wang et al. (2021b) who argued that glacier melt has increased its
contributions to the surface runoff by 12%-43% among the sub-basins of the Yarlung-Zangpo River basin (the mainstream
of Lhasa River) after 1997.
Some components of basin hydrology remain poorly observed, however we note that the combined increase in
precipitation and ablation detailed above was much notably greater than the observed increase in runoff especially in the
Yangbajain, a discrepancy that due to some combination of increased residential, industrial or agricultural water use
(Pritchard, 2019), increased evaporation (Han et al., 2021), and possible deep seeps in upper Lhasa River (Lin et al., 2020).
In the Nam Co basin, increase glacier runoff also appears to have been important in controlling the level of Nam Co
Lake. The Nam Co basin glacier mass balance ($0.32\pm0.16$ m w.e.a$^{-1}$) that we found for 1976-2014, equated to $30.9\pm15.4\%$
of the reported increase in Nam Co Lake water storage (Zhang et al., 2011b). This glacier contribution is comparable to

previous estimates of 52.9% for the 1971-2004 period (Zhu et al., 2009), 28.7% for 1999-2010 based on a mass balance of 0.59 m w.e.a$^{-1}$ (Zhadang glacier) (Lei et al., 2013), $10.50 \pm 9.00\%$ for 2003-2009 by based on a mass balance of $-0.27 \pm 0.13$ m w.e. a$^{-1}$ (Li & Lin (2017), and $17.5 \pm 7.6\%$ for 2000-2014 based on a mass balance of -0.32 m w.e. a$^{-1}$ (Ke et al., 2022). Differences in these contributions of glaciers to increases in lake level reflect differences in the time periods studied and variability in the rate of change in the lake. For example, Ke et al. (2022) reported that their average lake level change of $(0.26 \pm 0.04$ m a$^{-1}$ for 2000s–2014) is substantially higher than $0.14 \pm 0.18$ m a$^{-1}$ for 1994–2015 reported by Brun et al. (2020).

**5. Conclusions**

Based on KH-9, Landsat, SRTM and ASTER satellite data, we have quantified the changes of glacier area, surface elevation and mass balance in the WNR over the past 44 years and compared the effects of topography, debris-cover and proglacial lakes on glacier change during 1976-2000 and 2000-2020. Our major conclusions are:

(1) Glaciers in the WNR retreated by $295.73 \pm 43.45$km$^2$, or $33.42 \pm 4.9\%$ of their area, from 1976-2020, at a mean rate of $-0.76 \pm 0.11$ % a$^{-1}$. Over this time, they lost a total of $11.56 \pm 0.12$ Gt of ice. In addition, increased glacial meltwater in the WNR primarily drove increased runoff in Lhasa River basin and appeared to have been important in controlling the level of Nam Co Lake. The Nam Co basin glacier mass balance ($0.32 \pm 0.16$ m w.e.a$^{-1}$ during 1976-2014), equated to $30.9 \pm 15.4\%$ of the increase in Nam Co Lake water storage.

(2) The average retreat rate from 2000 to 2020 ($1.17 \pm 0.30$ % a$^{-1}$) was more than twice that from 1976 to 2000 ($0.54 \pm 0.21$ % a$^{-1}$). Similarly, the mean glacier mass balance from 2000 to 2020 ($-0.37 \pm 0.12$ m w.e.a$^{-1}$) was more negative than that from 1976 to 2000 ($-0.26 \pm 0.09$ m w.e.a$^{-1}$) (though the change is within the uncertainties). The more rapid ice loss from 2000 to 2020 was mainly due to intensified glacier melting after 2014, which was likely associated with particularly strong warming of the region after that year. Besides, areal retreat rate and mass loss rate of most glaciers was not synchronized during 1976-2000 and 2000-2020.

(3) In the WNR the spatial and temporal patterns of glacier loss can largely be explained by the observed patterns of regional climate change. Locally, the mass balance varied between different types of glaciers with proglacial lakes associated with the most rapid loss, particularly during 2000-2020. The mass balance of debris-covered glaciers was similar to debris-free glaciers during 2000-2020.

(4) Topographic setting influenced retreat and thinning, with loss rates decreasing with increasing elevation. The rate of both glacier retreat and thinning decreased with elevation, but the relationship between the parameters of slope and aspect with thinning rates differed from their relationship with retreat rates, spatially and through time. For slopes of 8-40° (which includes most glaciers), for example, the retreat rate increased with slope while the thinning rate decreased.

In this study, we observed accelerated glacier loss in the WNR on multi-year time scales. However, factors such as

precipitation, temperature and altitude could not yet fully explain the heterogeneity of glacier changes. Thus, more detail data and glacier ablation models are needed to fully understand the mechanism of glacier change in the future.

**Author contributions:**

Conceptualization, S.W. and J.L.; methodology, S.W.; software, S.W. and X.Q.; data curation, S.W., and J.Z; writing—original draft preparation, S.W.; writing—review and editing, H.P. and L.K.; visualization, H.P and J.L.; supervision, W.X. and Y.Z.; project administration, J.L.; funding acquisition, J.L.

**Acknowledgements:**

This work was supported by the Second Tibetan Plateau Scientific Expedition and Research Program (STEP; Ministry of Science and Technology, MOST; grant no. 2019QZKK0207), the National Natural Science Foundation of China (NSFC; grant no. 92047301).

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

Table 1 Glacier area changes over the WNR from 1976 to 2020

| | 1976 Area(km$^2$) | 2000 Area (km$^2$) | 2020 Area (km$^2$) | 1976-2000 △Area (% a$^{-1}$) | 2000-2020 △Area (% a$^{-1}$) | 1976-2020 △Area (% a$^{-1}$) |
|---|---|---|---|---|---|---|
| The WNR | 884.90±29.71 | 770.03±33.44 | 589.17±31.72 | -0.54± 0.21 | -1.17± 0.30 | -0.76± 0.11 |
| Lhasa River | 662.23±21.95 | 580.81±22.79 | 447.93±21.74 | -0.51± 0.20 | -1.14± 0.27 | -0.74± 11 |
| Nam Co basin | 222.58±7.76 | 189.22±7.62 | 141.22±7.10 | -0.62± 0.20 | -1.27± 0.32 | -0.83± 0.11 |

Table 2 Glacier elevation change, mass balance and total mass change over the WNR from 1976 to 2020

| | 1976-2000 | | | 2000-2020 | | |
|---|---|---|---|---|---|---|
| | Elevation change (m a$^{-1}$) | Mass Balance (m w.e.a$^{-1}$) | Total mass change (Gt a$^{-1}$) | Elevation change (m a$^{-1}$) | Mass Balance (m w.e.a$^{-1}$) | Total mass change (Gt a$^{-1}$) |
| The WNR | -0.31 ± 0.10 | −0.26± 0.09 | −0.24± 0.08 | −0.44± 0.13 | −0.37± 0.12 | −0.29± 0.09 |
| Lhasa River | -0.29± 0.12 | −0.25± 0.10 | −0.21± 0.10 | −0.40± 0.16 | −0.34± 0.14 | −0.26± 0.09 |
| Nam Co basin | -0.36 ± 0.17 | −0.31 ± 0.15 | −0.06± 0.02 | −0.52± 0.18 | −0.44± 0.16 | −0.06 ± 0.03 |

Table 3 Statistics of area, quantity, and mass balance of different types of glaciers

| Glacier type | 1976-2000 | | | 2000-2020 | | |
|---|---|---|---|---|---|---|
| | Area (km$^2$) | Number | Mass Balance (m w.e.a$^{-1}$) | Area (km$^2$) | Number | Mass Balance (m w.e.a$^{-1}$) |
| Lake-terminating glaciers | 70.29±2.69 | 46 | -0.36±0.26 | 49.6±1.82 | 34 | -0.56 ±0.31 |
| Debris-covered glaciers | 55.42±1.25 | 5 | -0.20±0.34 | 51.59±1.77 | 5 | -0.44 ±0.47 |
| Debris-covered and lake-terminating glaciers | 5.46±0.32 | 1 | -0.18±0.80 | 6.05±0.32 | 1 | -0.34±0.92 |
| Regular glaciers | 615.29±20.73 | 617 | -0.30±0.10 | 554.64±22.93 | 692 | -0.42±0.12 |


 **Figure captions**

Figure 1 (a) Overview of study area. (b) Glaciers distribution. Label I in the large, red dotted rectangle represents the SW
section of the WNR and Label II in the small, dark red dotted rectangle represents the NE section.
Figure 2   Glacier distribution in the WNR in 1976, 2000, 2009, 2014 and 2020. (a) Number and area of glaciers by size
category. (b) Distribution of glacier area with altitude. (c) Distribution of glacier area with slope. (d) Distribution of glacier
area with aspect. Data in 2009 came from Chinese Glacier Inventory II.
Figure 3   The distribution of glacier area change in the WNR from (a) 1976 to 2000, (b) from 2000 to2020, (c) 1976 to

668 2020.

Figure 4   Glacier area changes with (a) time, (b) elevation, (c) slope and (d) aspect. The short lines on either side of the point
indicate the margin of error in figure (a, b, c).
Figure 5    Mean annual glacier surface elevation changes in the WNR from (a) 1976 to 2000, (b) 2000 to 2020, and (c)
1976-2020. Label I in (a, b, c) represents the SW section and label II in (b) represents the NE section of the WNR (on
the same scale). The red rectangular box in (b) shows an area of the centra WNR referred to in the paper.
Figure 6   Glacier elevation change with altitude (m a.s.l) in the whole WNR, inside Nam Co drainage basin and outside Nam
Co drainage basin from (a) 1976 to 2000 and (b) 2000 to 2020. The dots represent the mean elevation change in each 50-m
elevation bin and shaded regions in the altitudinal distributions indicate the uncertainty.
Figure 7   The distribution of glacier-wide mass balance for individual glaciers (> 2 km$^2$) in the WNR from (a) 1976 to 2000,
and (b) from 2000 to2020. Label I represents the SW section and label II represents the NE section of the WNR (on the same
map scale).
Figure 8   Glacier elevation change from 1976 to 2000 and from 2000 to 2020 with (a) elevation, (b) slope, and (c) aspect. The
dots in figure (a) represent the mean elevation change in each 50-m bin and shaded region in (a) indicate the uncertainty in the
altitudinal distributions. (b) is boxplot of dh in 2-° slope bins and four lines from bottom to top for one box represent minimum
value, 25th percentile, 75th percentile, and maximum value, respectively and dots in figure (c) represent the mean elevation
change in each 2-° slope bin. (c) represent the mean elevation change in each 45-°aspect bin.
Figure 9 Glacier area changes with slope during 1976-2000 (a) and during 2000-2020 (b), and glacier elevation changes with
slope during 1976-2000 (c) and during 2000-2020 (d).
Figure 10    Rate of glacier elevation change with elevation of different glaciers types during (a) 1976-2000 and (b) 2000-2020.
Plots represent the mean values of glacier elevation change in each 50-m elevation bin and shaded regions indicate the
uncertainty in the altitudinal distributions.
Figure 11    (a) Glacier elevation change in the WNR during 2000-2014. (b) Glacier elevation changes with altitude in the
WNR, inside Nam Co drainage basin and outside Nam Co drainage basin from 2000 to 2014. The dots represent the mean
elevation change in each 50-m elevation bin and shaded regions indicate the uncertainty in the altitudinal distributions. (c)
Total area of glaciers and that area covered by the datasets during 1976-2000 and 2000-2014.
Figure 12 Temperature and precipitation changes for the study area at Damxung, Lhasa and Bange stations from 1976 to
2020. Annual average temperature and precipitation (a, b), ablation season (June to September) average temperature and
precipitation (c, d), accumulation season (January to May and October to December) average temperature and precipitation (e,
f).
Figure 13 Gridded temperature and precipitation change during specific time periods.

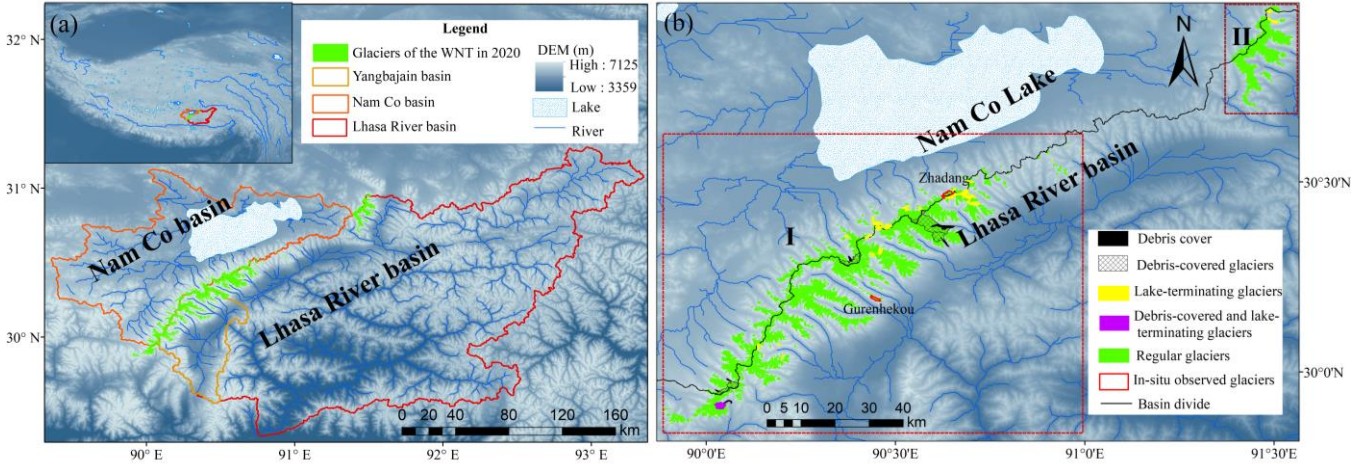


Figure 1 (a) Overview of study area. (b) Glaciers distribution. Label I in the large, red dotted rectangle represents the
SW section of the WNR and Label II in the small, dark red dotted rectangle represents the NE section.

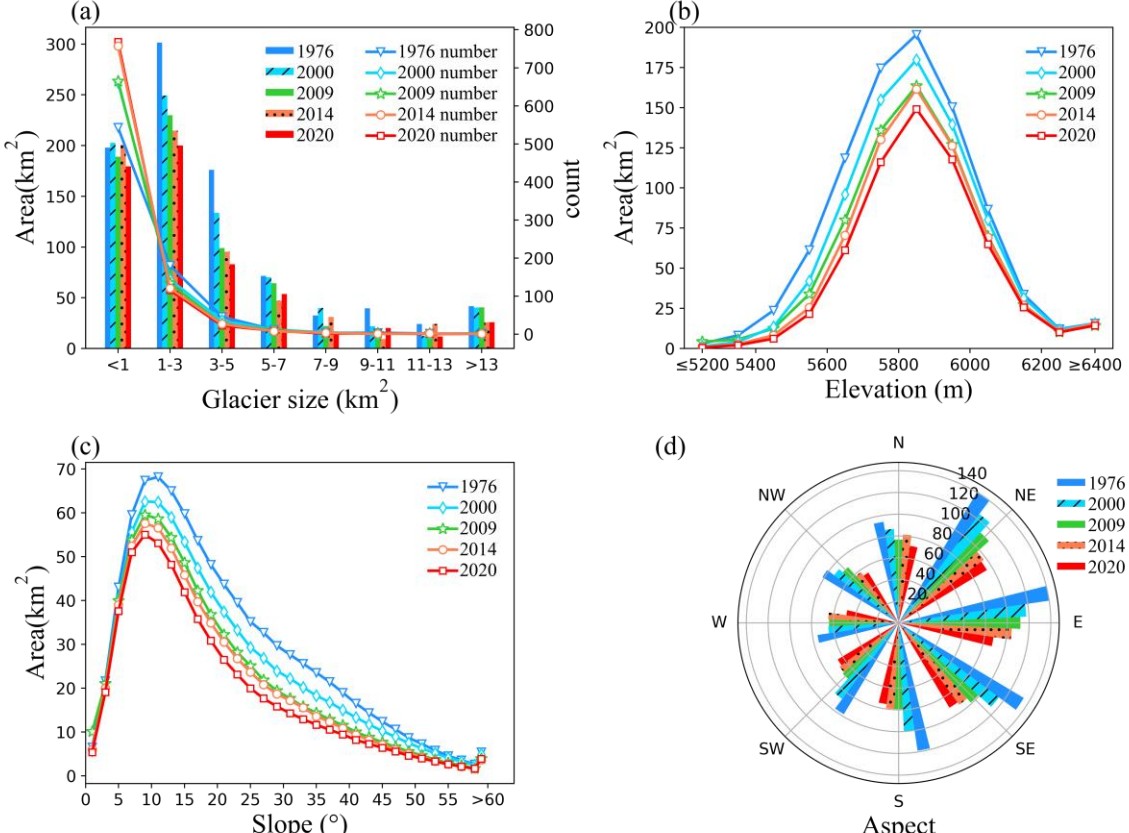

Figure 2    Glacier distribution in the WNR in 1976, 2000, 2009, 2014 and 2020. (a) Number and area of glaciers by size category. (b) Distribution of glacier area with altitude. (c) Distribution of glacier area with different slope. (d) Distribution of glacier area with aspect. Data in 2009 came from Chinese Glacier Inventory II.

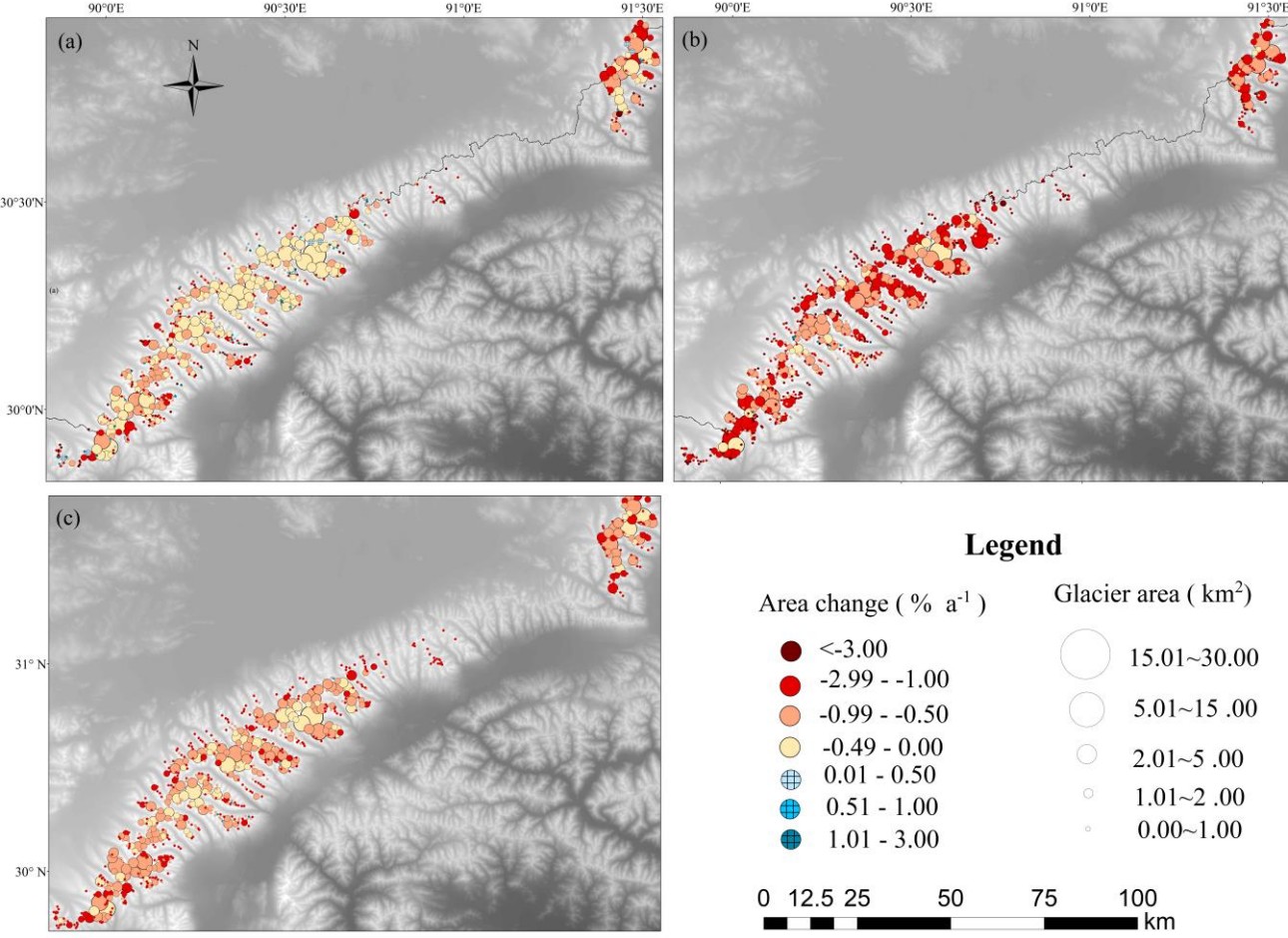

Figure 3    The distribution of glacier area change in the WNR from (a) 1976 to 2000, (b) from 2000 to 2020, (c) 1976 to
709    2020.

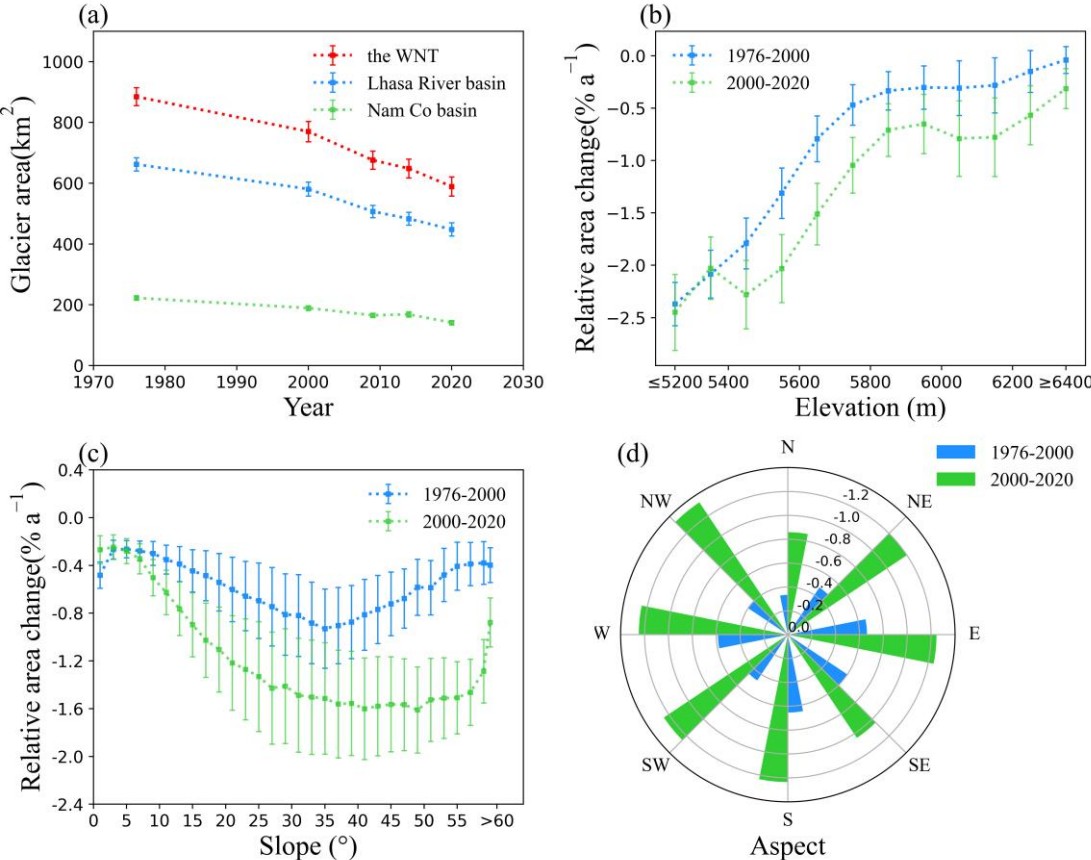


Figure 4     Glacier area changes with (a) time, (b) elevation, (c) slope and (d) aspect. The short lines on either side of the point
indicate the margin of error in figure (a, b, c).

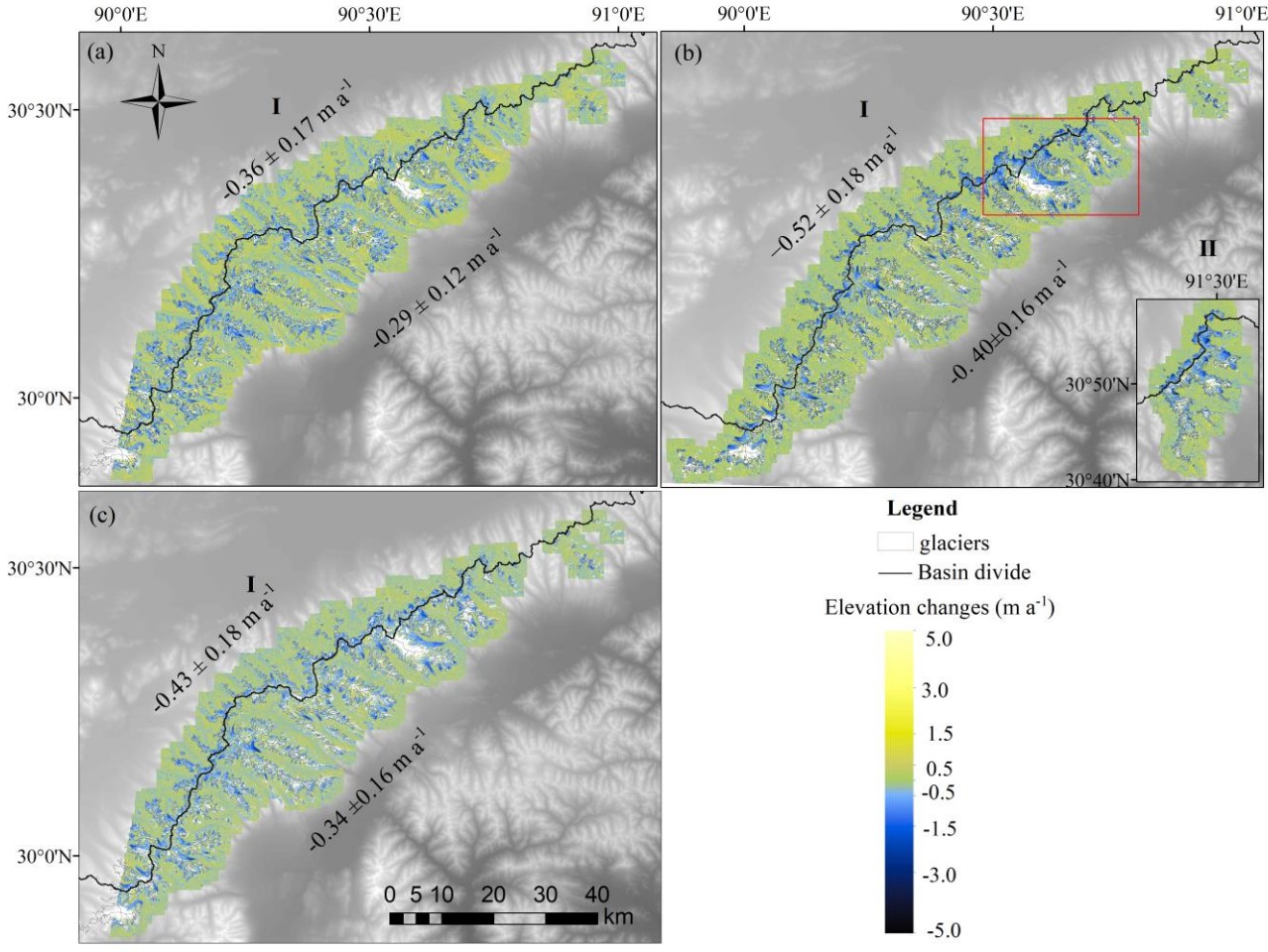


Figure 5    Mean annual glacier surface elevation changes in the WNR from (a) 1976 to 2000, (b) 2000 to 2020, and (c)

1976-2020. Label I in (a, b, c) represents the SW section and label II in (b) represents the NE section of the WNR (on

the same scale). The red rectangular box in (b) shows an area of the centra WNR referred to in the paper.

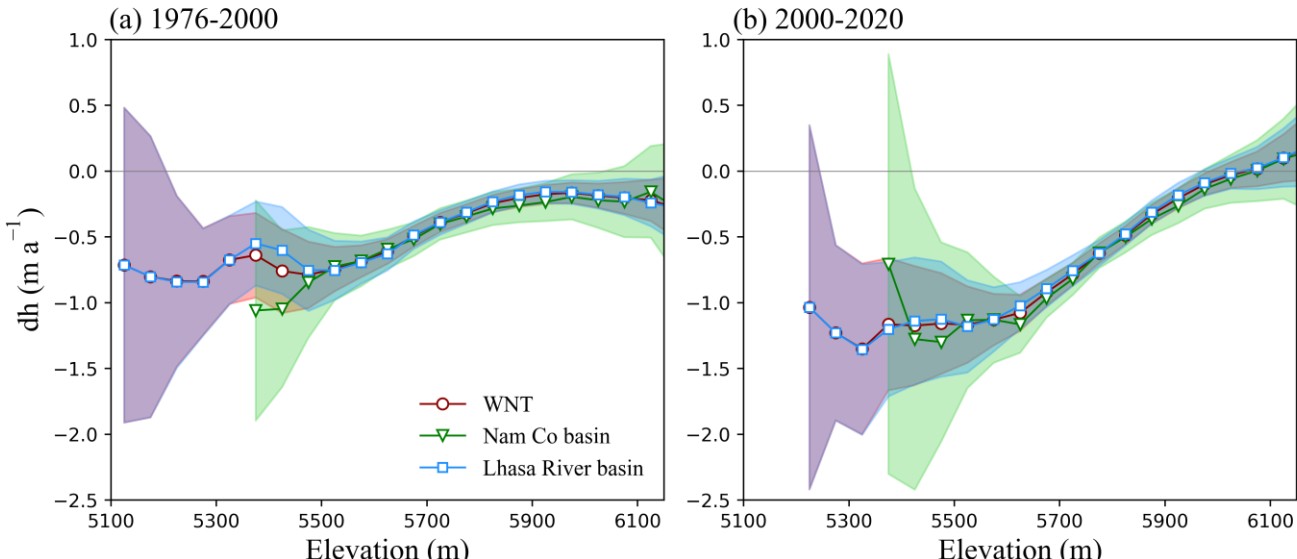


Figure 6 Glacier elevation changes in relation to elevation (m a.s.l) in the whole WNR, inside Nam Co drainage basin and
outside Nam Co drainage basin from (a) 1976 to 2000 and (b) 2000 to 2020 (b). The dots represent the mean elevation change
in each 50-m elevation bin and shaded regions in the altitudinal distributions indicate the uncertainty.

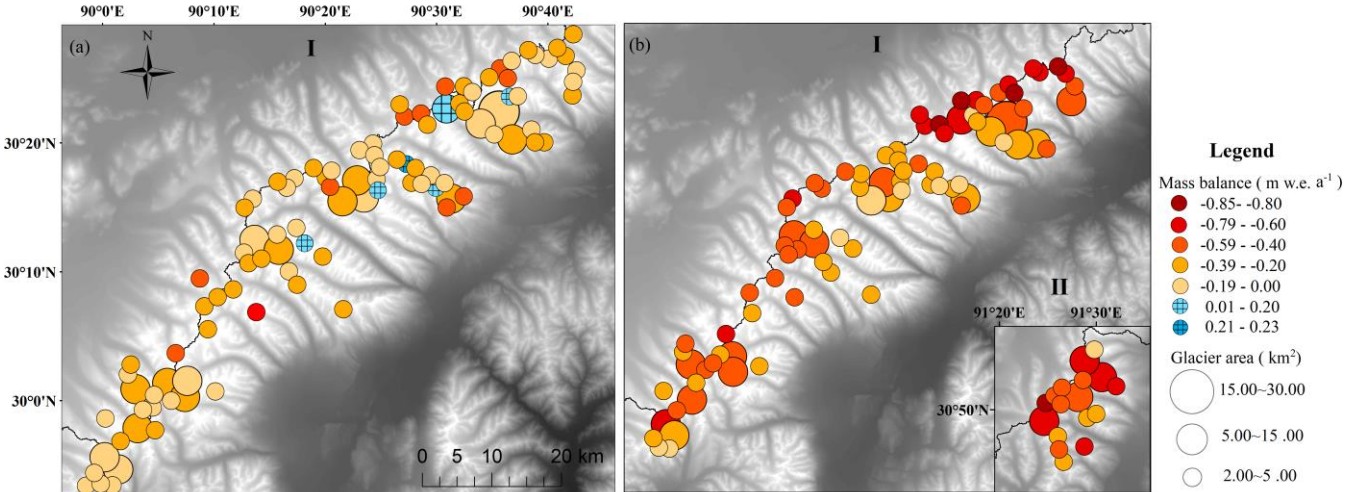


Figure 7 The distribution of glacier-wide mass balance for individual glaciers (> 2 km²) in the WNR from (a) 1976 to 2000
and (b) 2000 to 2020. Label I represents the SW section and label II represents the NE section of the WNR (on the same map
scale).

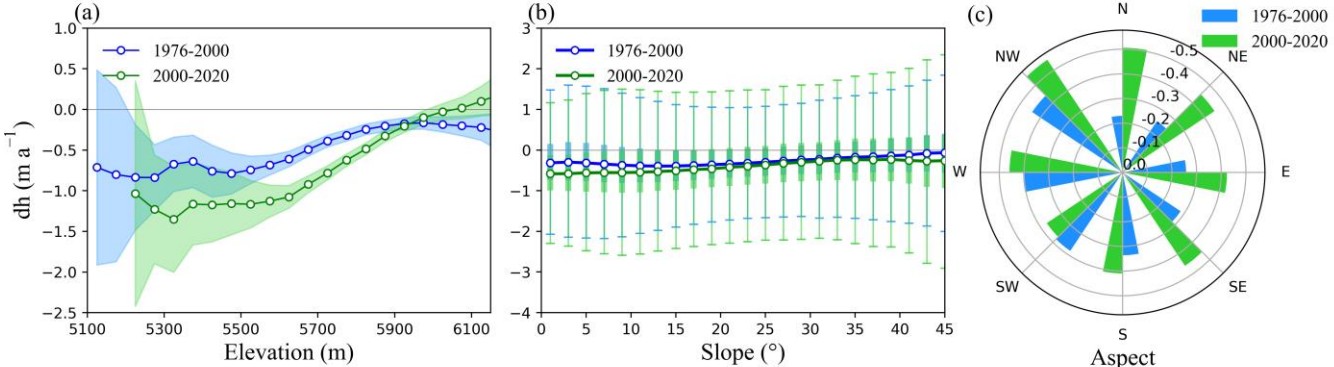


Figure 8 Glacier elevation changes from 1976 to 2000 and from 2000 to 2020 with (a) elevation, (b) slope and (c) aspect. The
dots in figure (a) represent the mean elevation change in each 50-m bin and shaded region in (a) indicate the uncertainty in the
altitudinal distributions. (b) is boxplot of dh in 2-° slope bins and four lines from bottom to top for one box represent minimum
value, 25th percentile, 75th percentile, and maximum value, respectively and dots in figure (c) represent the mean elevation
change in each 2-° slope bin. (c) represent the mean elevation change in each 45-°aspect bin.

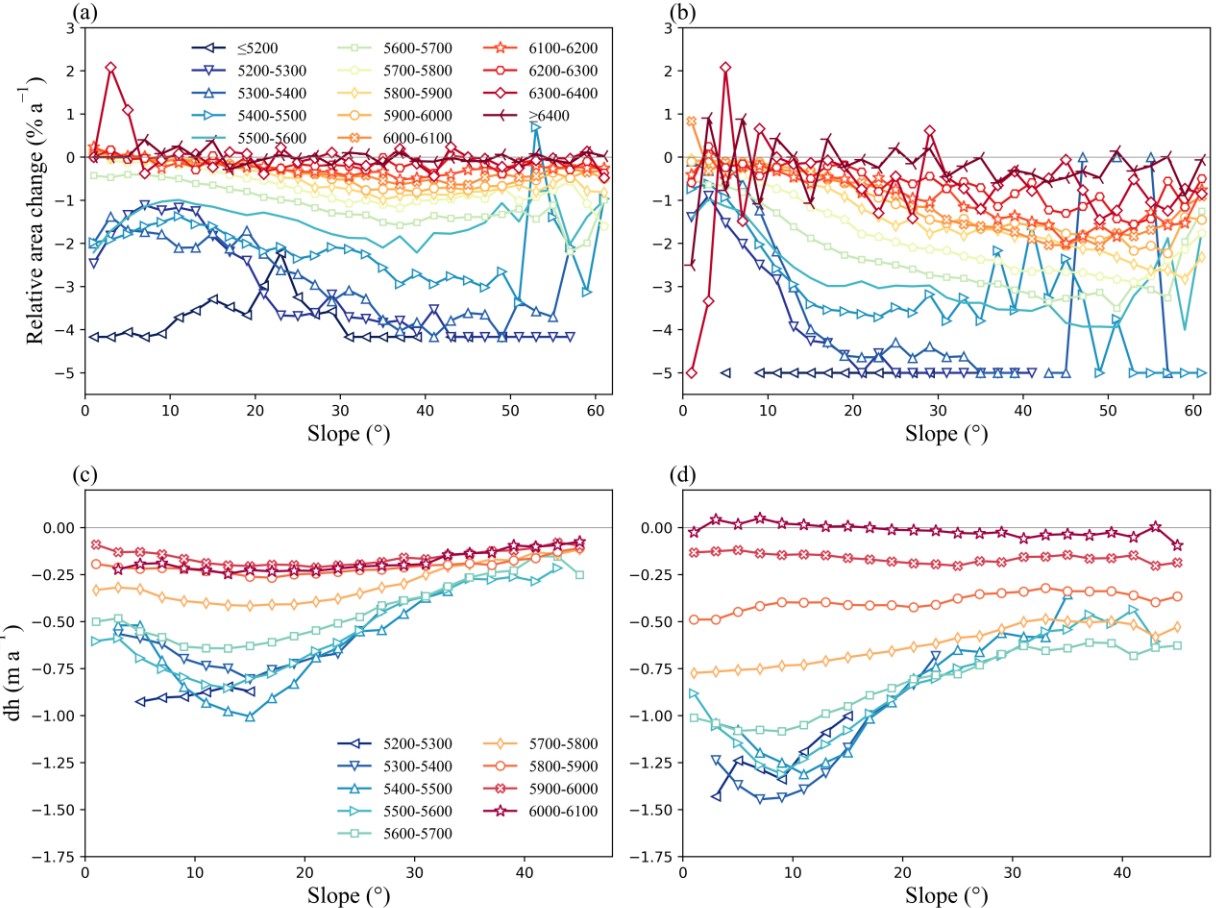

Figure 9 Glacier area changes with slope during 1976-2000 (a) and during 2000-2020 (b), and glacier elevation changes with slope during 1976-2000 (c) and during 2000-2020 (d).

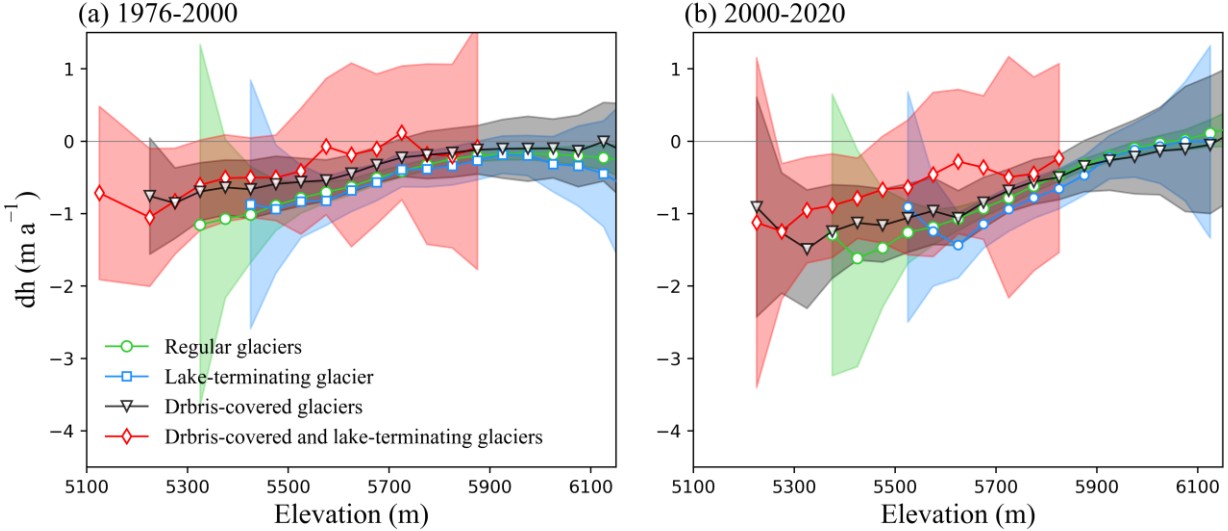

Figure 10    Rate of glacier elevation change with elevation of different type glaciers during (a) 1976-2000 and (b) 2000-2020 (b). Plots represent the mean values of glacier elevation change in each 50-m elevation bin and shaded regions indicate the uncertainty in the altitudinal distributions.

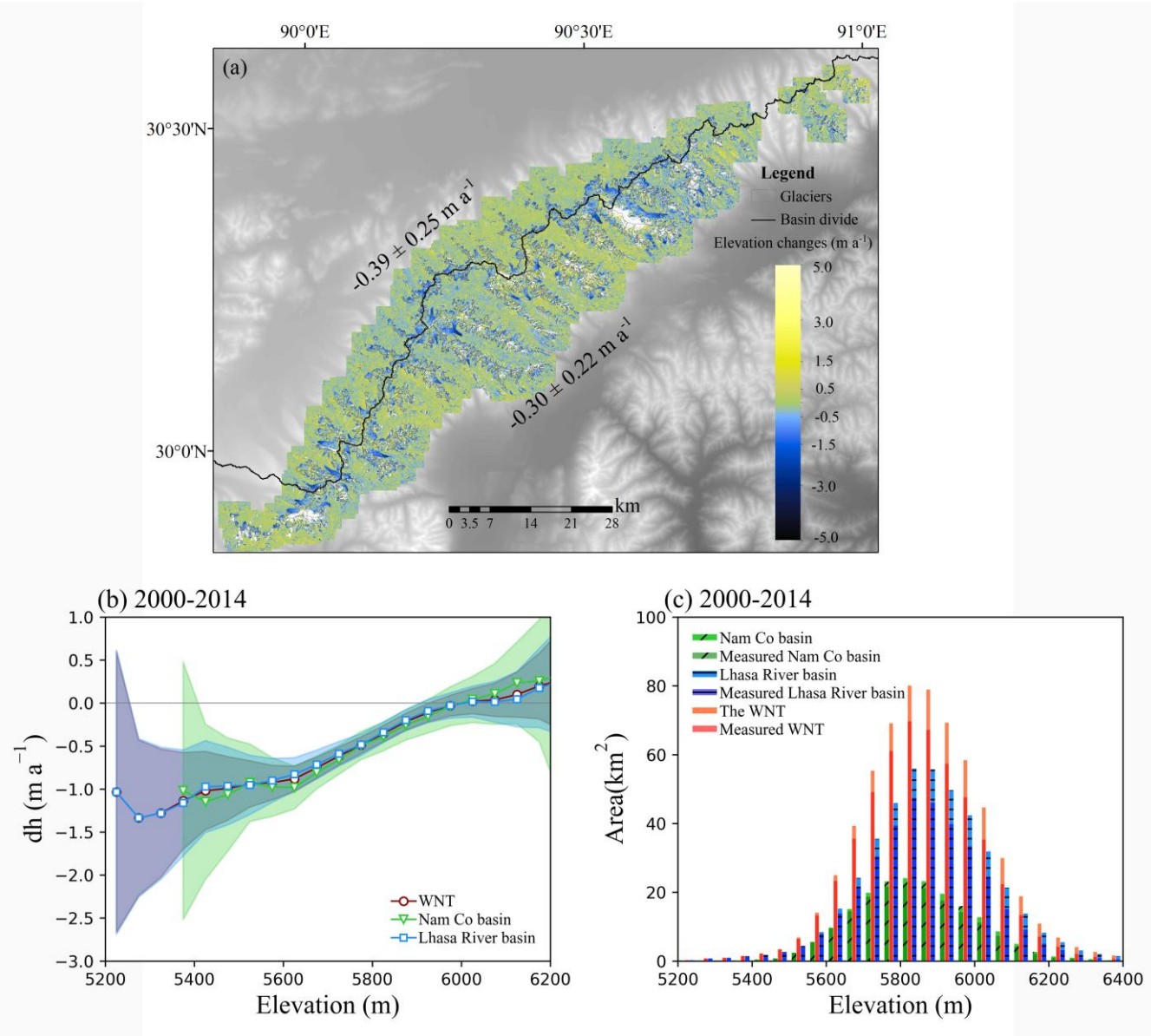


Figure 11     (a) Glacier elevation change in the WNR during 2000-2014. (b) Glacier elevation changes in relation to elevation
with altitude in the WNR, inside Nam Co drainage basin and outside Nam Co drainage basin from 2000 to 2014. The dots
represent the mean elevation change in each 50-m elevation bin and shaded regions indicate the uncertainty in the altitudinal
distributions. (c) Total area of glaciers and that area covered by the datasets during 1976-2000 and 2000-2014.

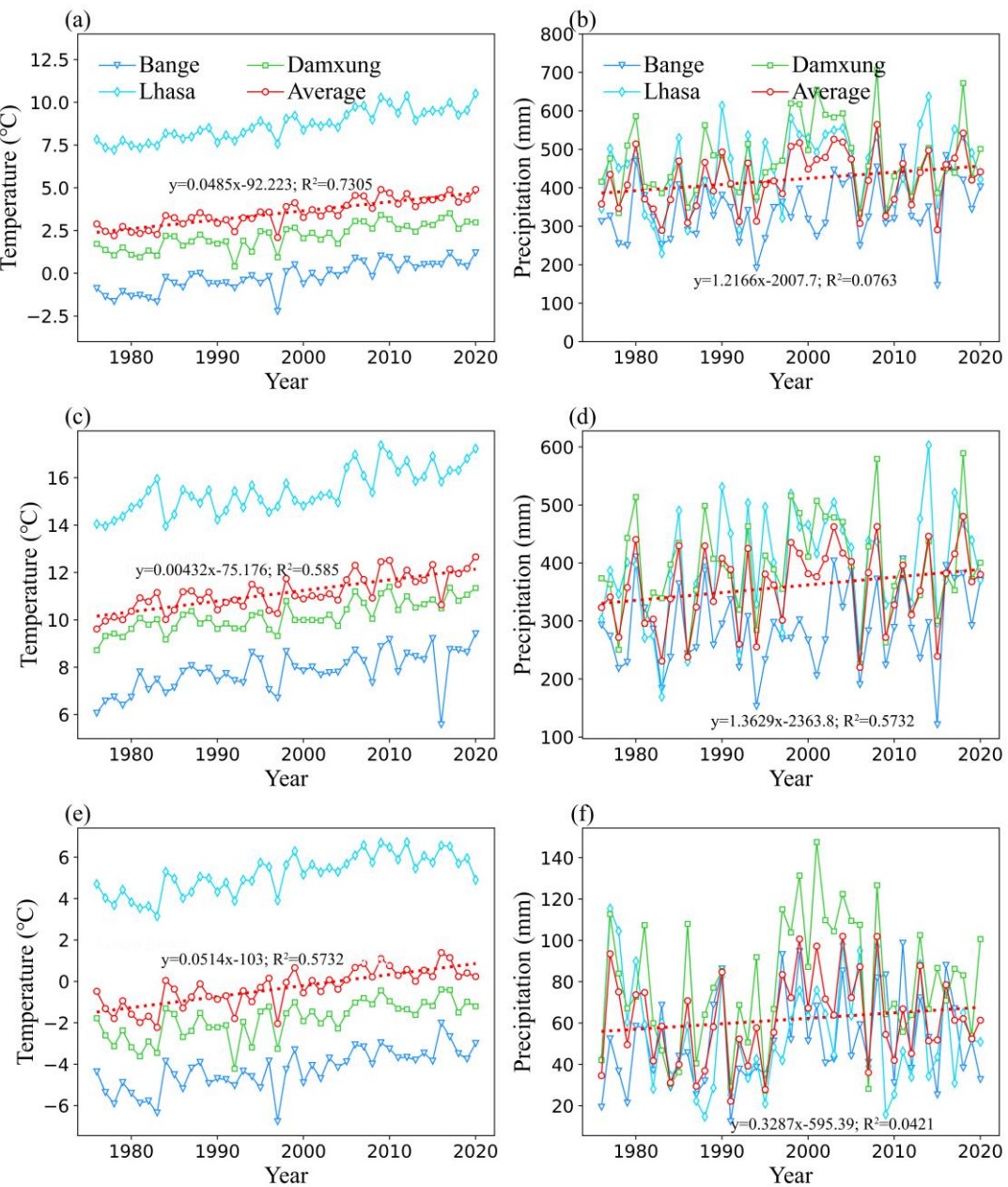


Figure 12 Temperature and precipitation changes for the study area at Damxung, Lhasa and Bange stations from1976 to
2020. Annual average temperature and precipitation (a, b), ablation season (June to September) average temperature and
precipitation (c, d), accumulation season (January to May and October to December) average temperature and precipitation (e,
f).


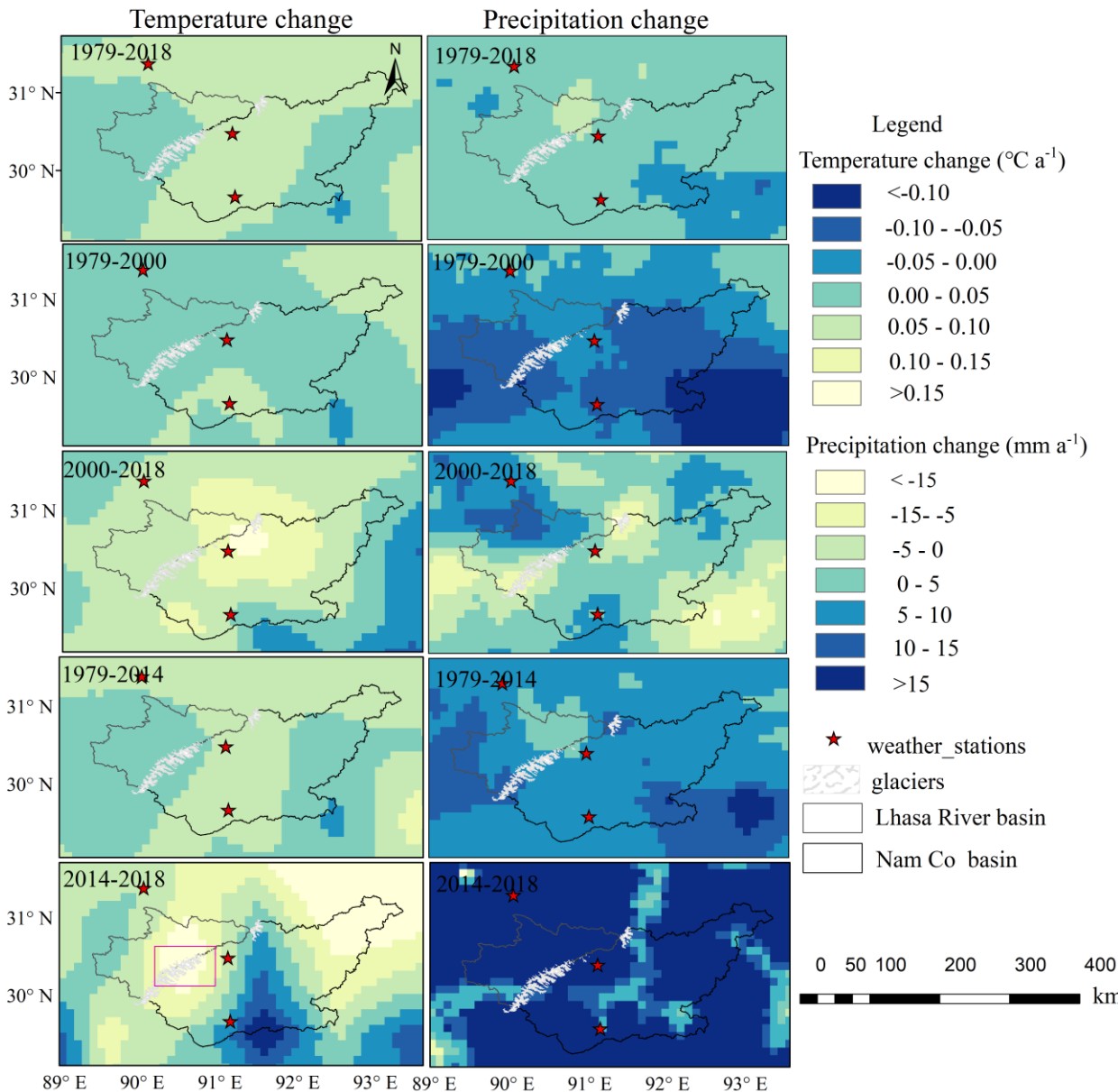


Figure 13    Gridded temperature and precipitation change during specific time periods.