# Peer review of "Characterizing four decades of accelerated glacial mass loss in the"

_Hydrology and Earth System Sciences, 2022_

## Author Response (AR1)

**Dear Editors and Reviewers,**

Many thanks for your valuable comments on our manuscript (hess-2022-179) titled "Characterizing four decades of accelerated glacial mass loss in the West Nyainqentanglha Range of the Tibetan Plateau." We have carefully addressed the reviewers and community's concerns and suggestions. The quality of this manuscript has been enhanced.

In the attached revised manuscript, blue colored texts represent what have been revised. Minor language corrections are not marked with blue color.

In the following (see Response to Referee #1, Response to Referee #2, and Response to community #1), we provide point-by-point response to each comment (blue texts are our responses, while black texts are original comments). Note that we have also modified the color scales for some figures to meet the journal's requirement.

Once again, we appreciate the time the reviewers put in reading our manuscript, and the comments were valuable, refreshing, and encouraging. My co-authors and I hope that we have adequately addressed all the review comments.

Sincerely Yours,

Jintao Liu, Shuhong Wang & Hamish D. Pritchard

**Response to Referee #1**

Hydrol. Earth Syst. Sci. Discuss., referee comment
RC1 https://doi.org/10.5194/hess-2022-179-RC1, 2022

[Figure]

**Comment on hess-2022-179**

Anonymous Referee #1
* * *
Referee comment on "Characterizing four decades of accelerated glacial mass loss in the West Nyainqentanglha Range of the Tibetan Plateau" by Shuhong Wang et al., Hydrol. Earth Syst. Sci. Discuss., https://doi.org/10.5194/hess-2022-179-RC1, 2022
* * *
The manuscript quantified changes in glacier area, surface elevation and mass balance in the WNT over the past 44 years and investigated associated influence factors over 1976-2000 and 2000-2020, based on multi-source remote sensing datasets. It is important to well understand the importance of glacier changes and associated impacts in the WNT, where these glaciers play a critical role in regulating regional water resources through supplying meltwater to the densely populated Lhasa River basin and Nam Co. Overall, the science of the manuscript is very interesting, and the structure and writing of the manuscript are good, but there are some issues the authors should be considered.

Response: Many thanks for the positive comments and suggestions. We have addressed the reviewer's concerns and suggestions carefully.

The key purpose of this study is to provide an internally consistent dataset of glacier area and mass change in the WNT over the past 44 years. What is your purpose for obtaining this dataset? It should be the hydrological impacts of glacier changes in the basin. However, there is no discussion on hydrological impacts of glacier changes on water resources of the basin or Nam Co, so the authors can consider some discussion about the influence of glacier change on hydrology in the WNT. It is very important for the manuscript, also for HESS.

Response: Many thanks for theses queries and suggestions. We agree that our motivation for compiling this dataset was to evaluate the hydrological effect of glacier changes on water resources downstream, and we have now added further information to section 2.2.6 (Hydrological Data) and 4.4 (Hydrological Effect) as follows.

Added to section 2.2.6:

In order to assess hydrological changes under glacier retreat, we have collected runoff data of the Lhasa River station during 1976-2013 and the Yangbajain station during 1979-2013 from the Tibet Autonomous Region Hydrology and Water Resources Survey Bureau.

We calculated the ratio of total glacier mass change to runoff in Lhasa River basin ($R_r$, %) and the total lake water storage change of Nam Co Lake ($R_l$, %) as follows:

$$R_r = \frac{\Delta M * S_g}{S_r R_a} \qquad (9)$$

$$R_l = \frac{\Delta M * S_g}{\Delta V} \qquad (10)$$

[revised manuscript text omitted]

Glacier outlines: Chinese Glacier Inventory (CGI) I and CGI II are available now. The authors generated new glacier boundaries of this region in the years of 1976, 2000, 2014 and 2020 from Landsat images obtained from various years. How about the differences between your results and previous datasets? What is the main reason why generate a new dataset? The authors may add some discussion or analysis in the manuscript or supplementary material.

Response:

The Chinese Glacier Inventory (CGI) I and CGI II of WNT represent extents in 1970 and 2009, so the time interval between the two periods of glacier inventories is relatively long. We wanted to show the process of glacier retreat under climate change. KH-9 (1976), SRTM (2000), and Aster Dems (2000-2020) are available in this area, this gave us a chance also to analyze the character of glacier thickness change for the 1976-2000 and 2000-2020 periods. Therefore, we extracted the areal extents of the same years 1976, 2000, 2020 to analyze changes in area and thickness together. Additionally, the glacier area in 2014 was extracted to test whether mass loss accelerated after 2014.

In section 4.1.1, we have now added a comparison of glacier area in our study with the Chinese Glacier Inventory (shown in Table S2). The CGI II of WNT in 2009 are in good agreement with the areal retreat trend in our study (also shown in Figure 2). The CGI I of WNT in 1970 is slightly smaller than the glacier area in 1976 in

our study, but it is within the margin of uncertainty. The CGI I was mapped based on the Chinese topographic maps, while glacier area in our study was mapped based on Landsat Images. The difference between them might come from this difference in data source used to extract the glaciers outlines. Besides, Frauenfelder & Kääb, (2009) reported that there are georeferencing errors in the areas in GGI I.

Table S2 Comparison of glacier area in this study with the Chinese Glacier Inventory

| | Glacier Area (km$^2$) | | | | | |
|---|---|---|---|---|---|---|
| | 1970 | 1976 | 2000 | 2009 | 2014 | 2020 |
| Chinese Glacier Inventory | 882.44 | - | - | 675.71 | - | - |
| This study | - | 884.90±29.71 | 770.03±33.44 | - | 648.55±30.88 | 589.17±31.72 |

Figure 2  Glacier distribution in the WNT in 1976, 2000, 2009, 2014 and 2020. (a) Number and area of glaciers by size category. (b) Distribution of glacier area with altitude. (c) Distribution of glacier area with slope. (d) Distribution of glacier area with aspect. Data in 2009 came from Chinese Glacier Inventory II.

Frauenfelder, R., & Kääb, A.: Glacier mapping from multi-temporal optical remote sensing data within the Brahmaputra River Basin, In Proceedings of the 33rd International Symposium on Remote Sensing of Environment, 4-8, 2009.

Meterological data: Please give the elevations of these meterological stations used in the manuscript.

Response:

There are three meteorological stations adjacent to the WNT, at Bange (31°23′N, 90°01′E, elevation of 4700 m), Lhasa (29°40′N, 91°08′E, elevation of 3648 m), and Damxung (30°29′N, 91°06′E, elevation of 4200 m).

We have added these details to Section 2.2.5. (Meteorological data) in the manuscript.

As shown in Table 3, the area of debris cover and lake terminating decreases between two periods, but thinning increases. Why? In particular, some current studies confirmed that the spatial expansion and thickening of the debris layer have been observed on different debris-covered glaciers with glacier shrinkage and sustained mass loss (e.g., Stokes et al., 2007; Kirkbride and Deline, 2013; Tielidze et al., 2020; Xie et al., 2020). Just as a matter of interest, what is the reason leading to the reduction of debris cover on glaciers of this region? In addition, between

two periods, glacier number increases from 617 and 692 with an area decreasing. What happened?

Response:

We are sorry for the mistake we made. The debris cover and lake terminating in Table 3 referred to debris-covered glaciers and lake-terminating glaciers. We have revised the Table 3 and the corresponding descriptions in the manuscript including, Figure 1 and 10.

The area of debris-covered glaciers and lake-terminating glaciers decreased, while surface lowering also accelerated, mainly driven by the continuous increase in temperature in the WNT region during 1976-2000, especially after 2014 (Figure 12 and Figure 13). In terms of the number and area of lake-termination, we identified glacier-marginal lakes as those lying within 50 m of a glacier boundary. As glaciers retreat, the distance between the end of the glacier and their proglacial lake increased, and some of lake-terminating glaciers in 1976 no longer belonged to lake-terminating class in 2000. This helps for explain the area decreased for this glacier class in Table 3.

For debris-covered glaciers, the area of debris cover actually increased from $6.60\pm1.15$ km$^2$ in 1976 to $7.37\pm1.48$ km$^2$ in 2020 in our study (Table S6, new added), and we note that this is not necessarily inconsistent with an overall glacier retreat. This is because increased melt rates that lead to surface lowering drive retreat of the glacier front, while also promoting a greater concentration of debris on the wider surface of glacier ablation area as more debris melts out from ice below. A spatial expansion of the debris layer has, for example, been observed on different debris-covered glaciers during retreat and sustained mass loss. (Stokes et al., 2007; Kirkbride & Deline, 2013; Tielidze et al., 2020; Xie et al., 2020). Unfortunately, no data are available to changes in the change of the thickness of the debris cover itself, and we assume that all glacier thickness changes resulted from loss of ice, without considering the thickness change of the debris cover layer. We think that this is reasonable in because in most area, debris layers are typically thin (order of 1 meter or less) and compared to elevation changes we map, and because most debris cover in the ablatio area emerge from englacial transport rather than direct deposition by new, local rock fall(e.g., McCarthy et al. 2017), so changes in the debris-layer thickness represent a redistribution of existing glacier volume, not a change in volume. We have now modified Section 4.2 (The influences of debris-cover and proglacial lakes on glacier mass changes) in the manuscript.

Table S6 Area changes of debris cover over the WNT from 1976 to 2020

| 1976 Area(km$^2$) | 2000 Area (km$^2$) | 2020 Area (km$^2$) | 1976-2000 △Area (% a$^{-1}$) | 2000-2020 △Area (% a$^{-1}$) | 1976-2020 △Area (% a$^{-1}$) |
|---|---|---|---|---|---|
| 6.60±1.15 | 6.90±1.34 | 7.37±1.49 | 0.20±1.12 | 0.28±1.45 | 0.24±0.65 |

The reason for the increased glacier number but decreased area is that intact glaciers break down into several smaller glaciers in the process of glacier ablation, e g., a large glacier in 1976 may become several smaller glaciers in 2020 (shown in Figure S3). We have added a comment on this to Section 3.1 (Glacier area change).

Kirkbride, M. P., & Deline, P.: The formation of supraglacial debris covers by primary dispersal from transverse englacial debris bands, Earth Surf Process Landf, 38(15), 1779-1792, 2013.

McCarthy, M., Pritchard, H., Willis, I. A. N., & King, E.: Ground-penetrating radar measurements of debris thickness on Lirung Glacier, Nepal, J Glaciol, 63(239), 543-555, 2017.

Stokes, C. R., Popovnin, V., Aleynikov, A., Gurney, S. D., & Shahgedanova, M.: Recent glacier retreat in the Caucasus Mountains, Russia, and associated increase in supraglacial debris cover and supra-/proglacial lake development, Ann. Glaciol, 46, 195-203, 2007.

Tielidze, L. G., Bolch, T., Wheate, R. D., Kutuzov, S. S., Lavrentiev, I. I., & Zemp, M.: Supra-glacial debris

cover changes in the Greater Caucasus from 1986 to 2014, Cryosphere, 14(2), 585-598, 2020.

Xie, Z., Haritashya, U. K., Asari, V. K., Young, B. W., Bishop, M. P., & Kargel, J. S.: GlacierNet: A deep-learning approach for debris-covered glacier mapping, IEEE Access, 8, 83495-83510, 2020.

[Figure]

Figure S3 Large glaciers break down into several smaller glaciers due to retreat. (a) Glaciers in Landsat MSS images from 1976-12-17. (b) Glaciers in Landsat 8/OLI images from 2020-09-29 (false-color composite of bands 7, 5, 4 for R, G, B).

The manuscript analyzed glacier area change and surface elevation change for the periods 1976-2000 and 2000-2020, how about the total changes in glacier area and surface elevation change between 1976-2020? The authors may add two figures in the manuscript or supplementary material that show changes between 1976-2020.

Response: We thank the reviewer for this suggestion, and we find that for 1976-2020, the mean glacier areal retreat rate in the WNT is -0.76± 0.11% $a^{-1}$ and surface lowering is -0.37 ±0.13 m $a^{-1}$ (equal to a water-equivalent loss rate of -0.31 ± 0.12 m w.e. $a^{-1}$ or a mass loss rate of -0.26 ± 0.09 Gt $a^{-1}$). We have added the area change of 2000-2020 in Figure 3, and surface change during 1976-2020 in Figure 5 and the corresponding description in line 235 and 256-257.

[Figure]

Figure 3 The distribution of glacier area change in the WNT from (a) 1976 to 2000, (b) from 2000 to2020, (c) 1976 to 2020.

[Figure]

Figure 5 Mean annual glacier surface elevation changes in the WNT from (a) 1976 to 2000, (b) 2000 to 2020, and (c) 1976-2020. Label I in (a, b, c) represents the SW section and label II in (b) represents the NE section of the WNT (on the same scale). The red rectangular box in (b) shows an area of the centra WNT referred to in the paper.

Minor comments:

1)Figure 1: Debris-cover is debris cover, Debris-cover glaciers is Debris-covered glaciers, and other glaciers is right?

Response: Actually, 'Other glacier' in Figure 1 corresponded to what we called 'Regular glaciers' in the text, and we have now corrected the terminology in Figure 1.

We have also corrected 'Debris-cover' to 'Debris cover', and 'Debris-cover glaciers' to 'Debris-covered glaciers'. We have also added an inset map to show the relative positions of the glaciers in the WNT, Lhasa River basin, and Nam Co basin. [Revised Figure 1].

[Figure]

Figure 1 (a) Overview of study area. (b) Glacier distribution. Label I in the large, red dotted rectangle represents the SW section of the WNT and Label II in the small, dark red dotted rectangle represents the NE section.

2) Some units should be superscript.

Response: Thanks, we have corrected these accordingly.

3) Some References cited in the manuscript are missing in the Reference list. Please carefully check.

Response: We have now checked these and added the missing references.

**Response to Referee #2**

Hydrol. Earth Syst. Sci. Discuss., referee comment RC3
RC1 https://doi.org/10.5194/hess-2022-179-RC3, 2022

[Figure]

**Comment on hess-2022-179**

Anonymous Referee #2
* * *
Referee comment on "Characterizing four decades of accelerated glacial mass loss in the West Nyainqentanglha Range of the Tibetan Plateau" by Shuhong Wang et al., Hydrol. Earth Syst. Sci. Discuss., https://doi.org/10.5194/hess-2022-179-RC3, 2022
* * *
This study assessed 44 years of glacier area and volume changes in the major West Nyainqentanglha Range (WNT) using comprehensive remote-sensed dataset. The selected study area is a very typical and important glacial region on the TP, bounded by the Nam Co basin to the north and the Lhasa River basin to the south. In addition to the widely-studied climate factors, the effect of local modulators, such as debris cover, slope and aspect, on glacier thickness has also been investigated. Overall, this study is very interesting and would merit publication in HESS.

Response: Many thanks for the positive comments and suggestions. We have addressed your concerns and suggestions carefully.

My comments are as following:

1. I am very interested in the impacts of elevation, slope, and aspect on the retreat rates and thinning rates. The elevation and slope may have correlations, so the contribution of each factor deserves further investigation. For example, one can do the partial correlation analysis or analyze the impact of slope in each elevation band.

Line 401: The following findings are interesting, and reasons need to be explained: "the retreat rate increased with slope while the thinning rate decreased."

Response: Many thanks for your suggestions. We have analyzed the impact of slope on glacier change in each elevation band (Figure 9). We found a positive correlation between areal retreat rates and slope (faster retreat with steeper slope) for most elevation bands and in both time periods (Figure 9 a and b). The only areas where this relationship differed were on flat or shallow slopes at lower altitudes (slopes below about 5° at elevations below about 5500 m, e.g., blue lines in Figure 9a) which also experienced relatively rapid retreat, and at the lowest elevations of <5200 m, with relatively few measurements available. We find a very similar but inverse relationship between slope and vertical thinning rates (dh in Figure 9 c and d). In this case, thinning rates were highest on shallow slopes and decreased over steeper slopes, except for flat or shallow slopes at lower altitudes where thinning rates were relatively low. We suspect that dominant pattern in which "the retreat rate increased with slope while the thinning rate decreased" occurred because:

    a) steep slopes are associated with thinner ice (Linsbauer et al., 2012). This means that any given thinning rate will tend to drive more rapid areal retreat on steeper slopes as the thinner ice there is depleted first, explaining the broadly positive correlation between retreat and slope; and

    b) steeper slopes are biased towards higher elevations, where the colder climate leads to slower thinning rates (dh), explaining the broadly negative correlation between slope and thinning rate.

The somewhat different behavior of the low-elevation flat areas (relatively rapid retreat, relatively slow thinning) may in part reflect the modulating effects of proglacial lakes (quicker retreat) and thicker debris cover (slower thinning) near the terminus.

We have added these details to Section 3.1 Glacier area change, 3.2 Geodetic mass balance and 4.3 Topographic and climatic controls of varying glacier mass loss in the manuscript.

[Figure]

Figure 9   Glacier area changes with slope during 1976-2000 (a) and during 2000-2020 (b), and glacier elevation changes with slope during 1976-2000 (c) and during 2000-2020 (d).

Linsbauer, A., Paul, F., & Haeberli, W.: Modeling glacier thickness distribution and bed topography over entire mountain ranges with GlabTop: Application of a fast and robust approach, Journal of Geophysical Research: Earth Surface, 117(F3), 2012.

2. Figure 1: the extent of the study area should be marked in the map of TP (the upper left small figure).

Response: We adjusted the Figure 1 when we responded the comments of the first referee and have marked the extent of the study area in the map of TP.

[Figure]

Figure 1 (a) Overview of study area. (b) Glacier distribution. Label I in the large, red dotted rectangle represents the SW section of the WNT and Label II in the small, dark red dotted rectangle represents the NE section.

3. Figure 5 is not easy to read. The legend of elevation changes and the boundaries of glaciers need to be adjusted.

Response: Thanks, we also adjusted the Figure 5 when we responded the comments of the first referee.

[Figure]

Figure 5 Mean annual glacier surface elevation changes in the WNT from (a) 1976 to 2000, (b) 2000 to 2020, and (c) 1976-2020. Label I in (a, b, c) represents the SW section and label II in (b) represents the NE section of the WNT (on the same scale). The red rectangular box in (b) shows an area of the centra WNT referred to in the paper.

4. It is difficult for me to understand the following sentences:

Line 309: "Compared to the glacier-area change between1970-2000 and 2000-2014 of Wu et al. (2016), and between1977-2000 and 2000-2010 of Wang et al. (2012), our results agree within the uncertainties over the whole WNT, and the southeast WNT respectively"

Line 323: "suggesting that the more strongly negative average for the longer 2000 to 2020 period (-0.37±0.12 m w.e.

a$^{-1}$) is the result of particularly strong negative mass balance after 2014"

Response: For line 309, what we are trying to say is that the deviation between our results and those from Wu et al. (2016) and Wang et al. (2012) over the whole WNT and the southeast WNT is within the margin of uncertainties.

For line323, we mean that the significantly more negative glacier mass balance from 2000 to 2020 is mainly due to the intensified glacier ablation after 2014. Because our glacier mass balance during 2000-2020 (-0.37±0.12 m w.e. a$^{-1}$) is more negative compared with the results during 2003-2009, 2000-2013/2014 from Li & Lin (2017), Neckel et al. (2014), Ren et al. (2020) and Zhang & Zhang (2017).We also calculated the change for the 2000-2014 period from ASTER DEMs (Figure 10) and found our estimated mass balance in this area (-0.28±0.15 m w.e. a$^{-1}$) is very similar to the other studies (Table S5). Therefore, we conclude that the significantly more negative glacier mass balance from 2000 to 2020 is mainly due to the intensified glacier ablation after 2014.This interpretation is supported by Ren et al. (2020) who also calculated a higher 2013-2020 thinning rate (-0.43±0.06 m w.e. a$^{-1}$) twice as negative as in 2000-2013.

We have made the corresponding supplementary description in line 306-307 and 319-321.

5. Some grammar and typo errors should be corrected, such as:

Line 101: WNT range mountain range

Line 311: our result. 4.1.2 Glacier mass balance.

Line 364 and 367: I cannot find Figure 3c and Figure 3d

Response: We are sorry for the mistake we made. We have corrected Line 101 and Line 311 in the manuscript. Figure 3c and Figure 3d should be Figure 4c and Figure 4d and we have checked all figure numbers in the manuscript.

**Response to community #1**

We thank Xiangying Li very much for the comments on our manuscript. We have addressed concerns and suggestions of Xiangying Li carefully. In the following, we provide point-by-point response to each comment (blue texts are our responses, while black texts are original comments).

1.The language is poor and should be revised and polished by a native English speaker at least.

Response: We think there might be some misunderstanding caused by different language habits. The manuscript has been fully revised two times by one of our co-authors named Hamish Pritchard, who is a native English speaker from British Antarctic Survey. He also has published many papers about glacier changes in many journals (e.g., nature; Front in Climates; Journal of Geophysical Research) and has a lot of experiences in scientific paper wring. We attached the versions of the manuscript revised by Hamish Pritchard with annotations.

2.Some terminology or expression should be corrected throughout the full text. For example, some should be glacial melt, glacial terminal, proglacial lake, changes in glacial area ...,

Response: We have read through the manuscript and checked the terms.

3.For the discussion on an increase in glacier populations as well as the response of authors "The reason for the increased glacier number but decreased area is that intact glaciers break down into several smaller glaciers in the process of glacier ablation". This is fully wrong and should be corrected throughout the full text because a glacier can change to 2 branches rather than 2 glaciers.

Response: The disagreement may come from two expressions of the same phenomenon. In the process of glacier ablation, one intact glacier breaks up into two parts, also counted by two glaciers in Chinese Glacier Inventory (CGI) I and CGI II. Some studies also reported that the area of glaciers decreases but the number of glaciers increased (Tielidze and Wheate 2018; Wu et al., 2016).

Tielidze, L. G., & Wheate, R. D.: The greater caucasus glacier inventory (Russia, Georgia and Azerbaijan), The Cryosphere, 12(1), 81-94.
Wu, K. Q., Liu, S. Y., Guo, W. Q., Wei, J. F., Xu, J. L., & Bao, W. J., et al.: Glacier change in the western Nyainqentanglha Range, Tibetan Plateau using historical maps and Landsat imagery: 1970-2014, J MT Sci Engl, 13(8), 1358–1374, https://doi.org/10.1007/s11629-016-3997-0, 2016.

4.Figure 1 is not clear and the classification for legend and glaciers is not easy to understand for readers. For example, in situ glaciers, others glaciers, .... In addition, some figures should be removed or merged into one figure.

Response: Thanks, we have revised the legend in Figure 1 as follows.

[Figure]

Figure 1 (a) Overview of study area. (b) Glacier distribution. Label I in the large, red dotted rectangle represents the SW section of the WNT and Label II in the small, dark red dotted rectangle represents the NE section.

5.I agree to the comments from RC1 "the authors can consider some discussion about the influence of glacier change on hydorology.... It is very important for the manuscript, also for HESS". This is extremely necessary for this study and should be a key point in the conclusions.

Response: We have added the impact of glacier ablation on hydrology when we responded the comments from RC1.

The part was also summarized in the conclusion. At the present stage, this journal only requires us to upload the document responding to the reviewer's comments, and the final revised manuscript should be uploaded later.

6.Relevant methods on glacier change can refer to published literature by some scholars. A lot of work has been done by Chinese scholars focusing on debris-covered glaciers (Haidong Han, et al.), proglacial lakes (Qiao Liu, Xin Wang, et al.), and changes in glacial area, elevation, mass balance, ... (Donghui Shangguan, Wanqin Guo, Shiyin Liu, et al.).

Response: We have added the impact of glacier ablation on hydrology when we responded the comments from RC1.